# Effects of cryo-EM cooling on structural ensembles

Lars V. Bock [1✉] & Helmut Grubmüller [1]

Structure determination by cryo electron microscopy (cryo-EM) provides information on structural heterogeneity and ensembles at atomic resolution. To obtain cryo-EM images of macromolecules, the samples are first rapidly cooled down to cryogenic temperatures. To what extent the structural ensemble is perturbed during cooling is currently unknown. Here, to quantify the effects of cooling, we combined continuum model calculations of the temperature drop, molecular dynamics simulations of a ribosome complex before and during cooling with kinetic models. Our results suggest that three effects markedly contribute to the narrowing of the structural ensembles: thermal contraction, reduced thermal motion within local potential wells, and the equilibration into lower free-energy conformations by overcoming separating free-energy barriers. During cooling, barrier heights below 10 kJ/mol were found to be overcome, which is expected to reduce B-factors in ensembles imaged by cryo-EM. Our approach now enables the quantification of the heterogeneity of room-temperature ensembles from cryo-EM structures.

[1] Theoretical and Computational Biophysics Department, Max Planck Institute for Multidisciplinary Sciences, Göttingen, Germany.
✉email: Lars.Bock@mpinat.mpg.de

Single-particle cryo-electron microscopy (cryo-EM) is a method to resolve 3-d structures of biomolecules. Over the past years, new electron detectors and improved image processing enabled cryo-EM maps at sufficient resolution to infer the atomic structure of a large range of macromolecules[1–3]. Today, atomic resolution can be achieved for single proteins, visualizing the densities of individual atoms[4,5].

In general, biomolecules perform their functions in solution. However, the direct study of specimens in liquid solutions using EM is impeded because the high vacuum required by EM is incompatible with the vapor pressure of liquid solutions. To preserve the biomolecules in the hydrated state, the sample is first spread into a thin film on a cryo-EM support grid. Subsequently, the grid is rapidly cooled, embedding the biomolecules in ice[6–10]. Because the formation of ice crystals would damage the sample, the cooling has to be rapid enough to result in vitreous (amorphous) ice[11,12]. Dubochet et al. provided an order-of-magnitude estimate for minimum cooling rate required for vitrification ($10^6$ K/s)[13], which would translate into cooling times shorter than 200 μs. Fast cooling rates are achieved by plunging the sample into the cryogenic liquid, most often liquid ethane kept close to its melting temperature of ~90 K. Alternatively, an ethane:propane mixture is used which has a lower melting temperature and remains liquid at 77 K presumably resulting in more rapid cooling[14]. An additional benefit of studying samples at cryogenic temperatures is the reduction of beam-induced radiation damage[15]. The radiation damage stems from energy deposited by electrons resulting in ionization of the sample and breakage of bonds which is reduced at lower temperatures. The vitrified sample is then transferred to a transmission electron microscope and the 2-d EM images of individual randomly oriented specimen are used to reconstruct 3D cryo-EM maps[16].

The rapid cooling during plunge-freezing preserves part of the structural heterogeneity generated by room-temperature structural fluctuations and, hence, contains information on functional motions[17–21]. However, cooling of the sample is expected to perturb the structural ensemble of biomolecules. In general, at room temperature, biomolecules can thermodynamically access more conformations and the rates for switching between conformations are faster than at low temperatures[22] (Fig. 1a). If the cooling is so rapid that essentially no barriers can be overcome in the process, the molecules are kinetically trapped in very nearby local minima and, hence, the room-temperature ensemble is preserved (Fig. 1a, ensemble after instant cooling). In contrast, if the cooling rate is very slow, the molecules spend more time at temperatures at which barriers can be overcome. The molecules are then more likely to equilibrate into the conformations thermodynamically accessible at the low temperature, i.e., the lower free-energy minima (Fig. 1a, slow gradual cooling). For intermediate cooling rates, one would expect that conformational changes with rates above a certain threshold (low barriers) would equilibrate into local free-energy minima during cooling. For conformational changes with slower rates (high barriers), the high-temperature ensemble would be largely preserved. For slow conformational changes where the high-temperature ensemble is preserved, the ensemble after cooling is expected to depend on the temperature before cooling[23]. An example for this scenario is given by cryo-EM reconstructions of ribosomes which were kept at different temperatures prior to cooling (37 °C, 18 °C, 4 °C)[17]. The 30S body rotation angle of ribosomes cooled down from a temperature of 37 °C showed a broad distribution. With lower temperature prior to cooling, the distribution narrowed, showing that information of the angle distribution is preserved during cooling. These differences of the distributions indicate that the 30S body rotation is too slow to equilibrate during the cooling process. The observation that captured conformations of a ketol-acid reductoisomerase and of temperature-sensitive TRP channels differ dramatically for different temperatures prior to cooling suggests that, in these cases, the minimal free-energy conformations depend on the temperature and that the conformations are preserved during rapid cooling[24–26]. Rapid cooling, with the freeze-quench method[27], is also used in electron paramagnetic resonance (EPR) spectroscopy experiments to trap intermediate states[28] and in combination with solid-state NMR experiments allows the identification of transient folding intermediates[29].

The effect that cooling down from an ensemble at lower temperature leads to a more homogeneous ensemble is used to obtain high-resolution cryo-EM structures by keeping the sample at temperatures below the physiological temperature before plunging, often at 4 °C (277.15 K)[4,5,30]. Further, the observation that some conformational changes are slower than the cooling rate is used in time-resolved cryo-EM which allows to obtain structural information for different time points of a system started out of equilibrium[31,32]. To that aim, e.g., after starting a reaction by mixing the reactants, the sample is frozen after different time intervals. Kinetic information can then be obtained from counting the specimens in the different states at different time points. The fastest accessible rates are determined by the shortest time interval from mixing to the frozen sample, currently reaching 6 ms[33].

The effects of low temperatures on protein dynamics and the coupling between solvent and proteins dynamics have been studied extensively using Mössbauer spectroscopy[34–37], x-ray crystallography[22,38–41], neutron scattering[42,43], IR spectroscopy[44], NMR[45], and molecular dynamics (MD) simulations[46–54]. The kinetic effects of cooling on biomolecules in crystals are expected to be very different from those embedded in thin solvent layers for single-particle cryo-EM. In contrast to the thin solvent layers, the crystal diameters are typically larger than 50 μm leading to cooling rates between 50 K/s and 700 K/s (ref. [55]), which is slower than the rates needed to vitrify pure water[41,56]. However, vitrification of water is achieved by added cryoprotectants[57] and by the highly concentrated proteins acting as cryoprotectants themselves[58–60]. Interestingly, the temperature dependence of the B-factors (Debye-Waller Factors, temperature factors) shows a glass transition between 180 K and 220 K from a smaller to a larger slope[34,36,40,42,43,46–51,53,61,62], e.g., from 1.2 Å$^2$ per 100 K to 6.4 Å$^2$ per 100 K (ref. [40]). In contrast to temperatures above the glass-transition temperature, crystallized ribonuclease-A cannot bind or unbind an inhibitor below the glass transition temperature indicating that the conformational change necessary for binding and unbinding is not accessible[39]. A dependency of the cooled ensemble on the cooling rate was hinted at by MD simulations of the protein carboxymyoglobin started from a single structure with very rapid cooling rates of 2 K/ps and 0.2 K/ps where a lower potential energy conformation was reached after the slower cooling, indicating that kinetics affect the cooled ensemble[50].

Here, we investigate the effects of plunge-freezing on structural ensembles by combining MD simulations with kinetic models of the cooling process. As a first step, for different water-layer thicknesses, we used a continuum model to estimate the temperature drop rates after plunging a water layer into liquid ethane. To probe how the cooling rate affects structural ensembles, we used explicit-solvent all-atom MD simulations of a ribosome·EF-Tu complex with decreasing temperatures at different rates (2 K/ps to 1.6 K/ns) to cool down an ensemble of 41 ribosome conformations. We chose the ribosome·EF-Tu complex because it includes very rigid parts, e.g., the surrounding of the peptidyl-transferase center, and large flexible parts which undergo conformational changes on 100-ns timescales, e.g., the L1 stalk[63]. Next, we trained and cross-validated kinetic models of the cooling process using the results from the MD simulations to estimate the effects of the cooling during plunge-

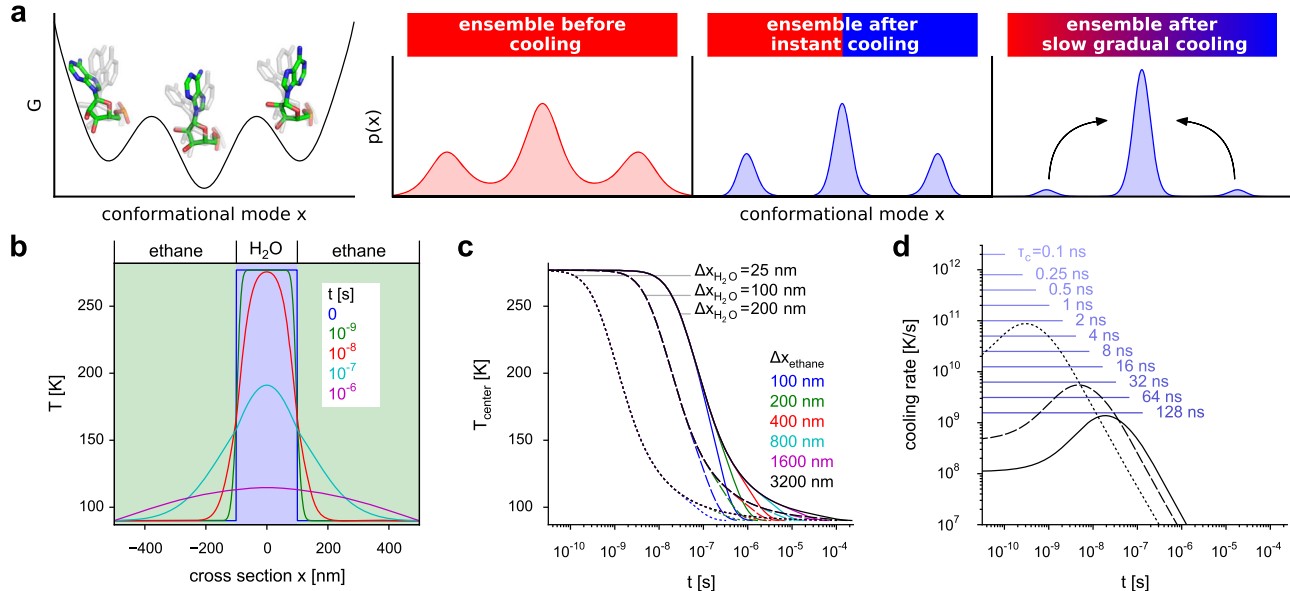

**Fig. 1 Effect of cooling on structural ensembles and estimated temperature drop during plunge-freezing. a** Schematic of a free-energy landscape along a conformational mode (left) and probability densities of structural ensembles (from left to right) before cooling, after instant cooling, and after slow cooling. **b** Temperature drop during plunge-freezing of a water layer embedded in ethane. Solution of the heat equation for a layer of water (thickness $\Delta x_{H_2O} = 200$ nm) surrounded by two layers of ethane (thickness $\Delta x_{ethane} = 400$ nm, each). The temperature profile $T$ is shown for different times $t$. Temperatures at the left and right borders are kept at 90 K. **c** The temperature $T_{center}$ at $x = 0$ nm is shown as function of time $t$ for different thicknesses of water and ethane layers. **d** The cooling rate at $x = 0$ nm is shown as function of $t$ for different water-layer thicknesses and for $\Delta x_{ethane} = 3200$ nm. Cooling rates for linear temperature decreases with different cooling time spans $\tau_c$ are shown as blue lines.

freezing. Finally, we combined the kinetic model with the temperature drops expected for different water-layer thicknesses used in cryo-EM experiments. This combination allowed us to predict the width of the structural ensemble, i.e., the B-factors before cooling from the B-factors observed in cryo-EM structures, thereby improving the interpretation of cryo-EM structures and their comparison to results from other experiments.

## Results

**Temperature drop during plunge-freezing**. The most common procedure to cool cryo-EM samples is to plunge the support grid with thin films of the sample into liquid ethane kept close to its melting temperature of 90 K. To estimate the rate and shape of the temperature drop within the sample, we considered a system consisting of three layers (Fig. 1b), one layer of water with a thickness of $\Delta x_{H_2O}$ surrounded by two layers of ethane with a thickness of $\Delta x_{ethane}$. The initial temperature profile at the time of plunging was set to 90 K for the ethane layers and 277.15 K for the water layer (Fig. 1b, blue line). To calculate the temperature profile evolution over time, the heat equation was solved numerically.

In experiments, different ice thicknesses between 15 nm and 200 nm depending on preparation and specimen were observed[5,8,64–66]. To estimate the effect of the water-layer thickness on the temperature drop, we used three values for $\Delta x_{H_2O}$, a minimal value of 25 nm slightly larger than the diameter of the ribosome and two values of 100 nm and 200 nm which capture the range of observed thicknesses. In experiments, the water-layer width is orders of magnitude smaller than the width of the ethane container. As a consequence, the water layer reaches temperatures below the glass transition temperature before the temperature increase in the ethane layer reaches the walls of the container. Therefore the continuum model does not need to include the whole ethane container and it suffices to make $\Delta x_{ethane}$ large enough such that the temperature drop in the water

layer is not affected by increasing $\Delta x_{ethane}$. To test which ethane-layer width is sufficient, we successively increased $\Delta x_{ethane}$ from 100 nm to 3200 nm. Figure 1c shows the temperature at the center of the water layer $T_{center}$ as a function of time. For small values of $\Delta x_{ethane}$, the temperature drops more rapidly, because temperatures larger than 90 K reach the outer boundaries earlier. With the largest $\Delta x_{ethane}$ values, this deviation occurs when the temperature is close to the ethane temperature (<100 K), indicating that for $\Delta x_{ethane} = 3200$ nm the effect of the boundaries on the relevant part of the cooling process is small and therefore will not be considered further.

For the thinnest water layer (25 nm), temperatures below 150 K are reached within 4 ns, whereas for the thicker layers (100 nm, 200 nm) it takes 64 ns and 250 ns, respectively. Temperatures below 100 K are reached within 120 ns, 1.9 μs and 7.6 μs. Before reaching 150 K, for water thicknesses between 25 nm and 200 nm, cooling rates in the range between $10^8$ K/s and $10^{11}$ K/s are observed (Fig. 1d). We therefore decided to use cooling rates between $2 \times 10^9$ K/s and $2 \times 10^{12}$ K/s (blue lines) for the subsequent temperature-quench (T-quench) MD simulations. As can be seen, the slowest cooling rates capture water thicknesses between 100 nm and 200 nm, whereas the faster cooling rates describe systems with minimal water layer thickness. Further, these cooling rates are consistent with the estimated minimum cooling rate of $10^6$ K/s required for vitrification[13].

Apart from the water-layer thickness which can be measured using tomography[66], the temperature drop also depends on the position within the layer with the slowest drop in the center (Fig. 1b). This is relevant, because in the time between the spreading of the sample onto the grid and the plunging, the biomolecules tend to adsorb to the air-water interface[67].

**Effects of different cooling rates on the ensemble**. To quantify the effects of different cooling rates on structural ensembles of large biomolecular complexes typically studied by cryo-EM, we used T-quench simulations, i.e., all-atom explicit-solvent

MD simulations of a ribosome · EF-Tu complex with linearly decreasing temperatures at different rates. A similar approach was proposed in a recent viewpoint article[68]. To that aim, we repeated T-quench simulations 41 times for each cooling rate, starting from 41 snapshots of an ensemble at a temperature of 277.15 K (4 °C). At this temperature, the samples were prepared prior to plunge-freezing in the cryo-EM experiment that resolved the ribosome · EF-Tu complex[30]. The latter ensemble was generated from a 3.5-μs MD simulation started from the cryo-EM structure. During the simulation, the deviation from the starting structure measured by the root mean square deviation (rmsd) approaches 5 Å (Fig. S1a), which is similar to the rmsd obtained from 2-μs simulations of the same system at 300 K presented earlier[69]. The observation that, at 300 K, similar rmsd values are reached in shorter time suggests that the dynamics on the timescales of the simulations is markedly slowed down by the lower temperature, as expected.

During cooling, one would expect that the structural heterogeneity of biomolecules decreases as the biomolecules equilibrate into local free-energy minima at the lower temperatures. As a result, the broader room-temperature ensemble should become narrower, depending on whether or not the relevant energy barriers can be overcome during the cooling process. To quantify this process, we recorded the distribution of root mean square fluctuations (rmsf) of atoms during the T-quench simulations. The rmsf values correspond to the width of the distribution of atom positions and therefore serve as a measure of structural heterogeneity. As a reference, an ensemble of structures that is sufficiently converged with respect to this observable was required. To that aim, we extracted structures of the ribosome complex, which contain all atoms resolved in the cryo-EM structure[30], at intervals of 50 ns from the 3.5-μs simulation at 277.15 K (Fig. S1a). Then, we grouped the 41 structures from the time points between 0 μs and 2 μs into an ensemble, aligned the structures, and then calculated the rmsf of each atom with respect to the average structure of the ensemble. This rmsf calculation was repeated for intervals 0.1–2.1 μs, 0.2–2.2 μs, …, and 1.5–3.5 μs and the resulting rmsf distributions are shown in Fig. S1b.

Throughout this work, we will quantify the structural heterogeneity of an ensemble with the 6-quantiles of the rmsf distributions, i.e., the 5 rmsf values Q1 to Q5 that divide the set of rmsf values into subsets of equal size. The rmsf distributions for the different time intervals show that initially the rmsf values decrease with increasing simulation time until 0.3–2.3 μs (Fig. S1b). For later time intervals, the rmsf values increase with a decreasing slope (intervals 0.4–2.4 μs to 1.5–3.5 μs). The observed behavior is consistent with a slight adaptation of the initial ribosome structure obtained at cryogenic temperatures to near room temperature, with subsequent room-temperature fluctuations. For the T-quench simulations, we expected a decrease in rmsf values. To observe this effect, it is crucial to start cooling from an ensemble of structures that does not show a decrease in rmsf values in the absence of cooling. We therefore chose the ensemble of 41 structures in the 1–3 μs interval (Fig. S1b, red histogram). The small rmsf increase for later time intervals does not significantly affect the results of the T-quench simulations (for details, see Supplementary Results).

To address the question of how the width of the ensemble, measured by the rmsf distribution, depends on the cooling rate, we started T-quench simulations from the 41 structures of the 277.15 K ensemble (Fig. 2a, red points) with the 11 different cooling time spans $\tau_c$ (0.1, 0.25, 0.5, 1, 2, 4, 6, 16, 32, 64, and 128 ns) that capture the plunge-freezing cooling rate (Fig. 1d). For each of the starting structures and for each cooling time span $\tau_c$, an MD simulation of length $\tau_c$ was started during which the

temperature was linearly decreased to reach 77 K at the end of the simulation.

For each cooling time span, we then extracted the final structures of each of the 41 simulations and calculated the rmsf distribution (Fig. 2b, blue histograms). Indeed, the observation that the quantiles from the cooled ensembles (blue lines) are consistently lower than those from the 277.15 K ensemble (red lines) indicates that, first, the cooling on the tested timescales reduces the heterogeneity of the structural ensemble, and, second, that the number of performed simulations is large enough to observe this effect.

The ribosome-complex is very heterogeneous, comprising RNA molecules and proteins, which allows us to test if the rmsf decrease differs for different parts. As expected, the absolute rmsf values before and after cooling are very different with protein atoms having larger values than RNA atoms (Fig. S2a). Further, the rmsf values increase with increasing distance from the center of the ribosome. However, the rmsf decrease during the T-quench simulations is remarkably similar for these subsets of atoms (Fig. S2b). Therefore, from here on we will use the rmsf distributions of all atoms to characterize the effects of cooling on structural ensembles.

To obtain rmsf values during the course of T-quenching, for each of the cooling time spans $\tau_c$, we first extracted conformations from all 41 simulations at 11 time points (0 ns, $0.1\tau_c$, $0.2\tau_c$, …, $\tau_c$). Next, for each cooling time span and time point, the rmsf values of all atoms were calculated from the 41 conformations. The 6-quantiles of these rmsf distributions are shown in Fig. 2c (Q1–Q5, cyan lines). The light cyan areas denote the standard deviations of the rmsf values obtained from bootstrapping of the conformations.

In addition to the decrease in heterogeneity, we observed a decrease in the size of the ribosome over the course of the cooling simulations, reflecting simple thermal expansion. To quantify this decrease, we applied a scaling factor $s$ to all of the extracted coordinates (relative to the center of geometry). For each conformation, the scaling factor $s$ that minimized the rmsd from the corresponding starting structure was calculated. For all cooling time spans, we saw the expected linear increase of $s$ as a function of the cooling time (Fig. S3). The scaling factors correspond to a thermal linear expansion coefficient between $4 \times 10^{-5}\,K^{-1}$ and $6 \times 10^{-5}\,K^{-1}$ and a volume decrease between 2.5 % and 3.3 % at the end of the cooling trajectories. This size decrease is in agreement with a volume reduction by 3 % observed in previous MD simulations after switching the temperature from 303 K to 85 K[54]. Further, from crystal structures of ribonuclease-A obtained at nine temperatures from 98 K to 320 K (ref. [40]), we calculated a linear expansion coefficient of $4 \times 10^{-5}\,K^{-1}$ (standard deviation $1 \times 10^{-5}\,K^{-1}$). Assuming that the heat expansion coefficient is rather independent of the system size, this good agreement suggests that our simulations describe the effect of cooling on the structural ensemble accurately.

To test if the observed changes in the rmsf distributions are largely a result of the temperature-induced shrinkage of the whole ribosome or, if not, to what extent the shrinkage affects the rmsf distributions, we scaled each conformation, which was extracted from the cooling trajectories, with the corresponding value of $s$. For each cooling time span and each time point, the rmsf quantiles were calculated before and after rescaling. As shown in Fig. 2c, the rescaling results in increased rmsf values, but it does not fully compensate the drop in rmsf values during cooling. This result suggests that in addition to the decrease in size, cooling does affect the heterogeneity of the structural ensembles even for very rapid cooling.

To check if the protein backbone and side-chain conformations are affected by the cooling, we extracted Φ and Ψ as well as $\chi_1$ and $\chi_2$ angles from the ensembles before cooling, as well as during and

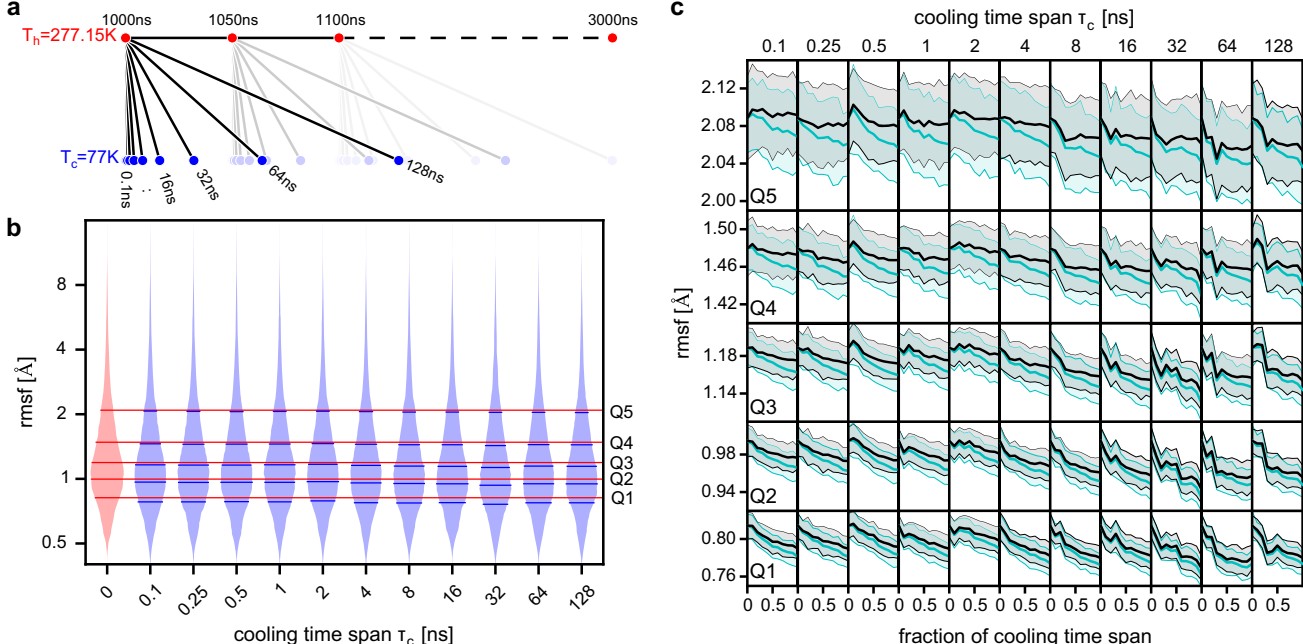

**Fig. 2 Effects of different cooling rates in T-quench MD simulations. a** Schematic of T-quench simulation protocol. From a trajectory of the ribosome · EF-Tu complex at $T = 277.15$ K, 41 snapshots were extracted (1000–3000 ns, every 50 ns). From these snapshots, T-quench simulations of different lengths (cooling time span $\tau_c$ 0.1 ns to 128 ns) were started with linearly decreasing temperature from 277.15 K to 77 K. **b** Histograms of the root mean square fluctuations (rmsf) of the heavy atoms of the ribosome · EF-Tu complex obtained from ensembles of 41 snapshots. The histogram for the ensemble before cooling is shown (light red area). Red horizontal lines show the 6-quantiles (Q1–Q5). For each cooling time span $\tau_c$, the rmsf histogram of the ensemble of the final snapshots of the 41 cooling simulations is shown (light blue area) with 6-quantiles (horizontal blue lines). **c** Rmsf quantiles are shown as a function of the simulation time before (mean value: cyan line, standard deviation: light cyan area) and after rescaling the conformations (black line, gray area).

at the end of all the T-quench simulations. The distribution of angles gets markedly narrower upon cooling which is quantified by decreasing Shannon entropies (Fig. S4). Interestingly, slower cooling results in more pronounced entropy decreases suggesting that some free-energy barriers which govern protein backbone and side-chain conformational changes can only be overcome during slower cooling.

**Thermodynamic and kinetic models of the cooling process**. During the T-quench simulations, the structural heterogeneity decreases for all cooling time spans, as indicated by the quantiles of the rmsf distributions (Fig. 2c). In addition, the decrease appears to be more pronounced with increasing cooling time spans, i.e., with slower cooling rates, which would indicate a kinetic contribution to the decrease. To quantify the effects of cooling and to separate thermodynamic (cooling-rate independent) from kinetic (rate dependent) contributions to the decrease, we used the results from the T-quench simulations to train and validate three different models of the cooling process. First, a thermodynamic model that does not depend on the cooling rate (Fig. 3a, model1) and two kinetic models that do depend on the cooling rate (Fig. 3b, c; model2 and model3).

The cooling-rate independent model1 describes the rmsf of an ensemble of atoms that can be trapped in different positions without the possibility to switch between the conformations during cooling. In this model, the atoms are subjected to harmonic potentials with a force constant $c$. The minima of the potentials are uniformly distributed in an interval from $-d$ to $d$ (Fig. 3a). The parameter $d$ determines the spread of the different accessible positions of the atoms.

To test for cooling-rate dependent effects, we used an established kinetic two-state model that was used previously to

study of the dynamic behavior of proteins at different temperatures (Fig. 3b; model2)[35,41,42]. Here, states A and B are located at a distance $\Delta x$ and the free energy of state B is larger than that of state A by $\Delta G$. The rates of switching between the states $k_{AB}$ and $k_{BA}$ are governed by the barrier height $\Delta G^{\ddagger}$ and the temperature (see Methods). The pre-exponential factor of the modified Arrhenius equation is $\kappa(T(t)/T_h)^{\nu}$, where $T(t)$ is the temperature as a function of time, $T_h$ the temperature before cooling, and $\kappa$ is a scaling factor ($\kappa = 1$ ns$^{-1}$, see Supplementary Results). The temperature exponent $\nu$ controls how the temperature enters in the pre-exponential factor.

Model3 is a combination of model1 and model2, where, in addition to the two-state kinetics of model2, the probability distribution in the two states is governed by uniformly distributed harmonic potentials (Fig. 3c). This model can describe the kinetic effects of a barrier between states that results in cooling-rate dependent behavior and the equilibration in multiple conformations.

To obtain probability distributions of the parameters for each model, we used Bayes' theorem and Metropolis sampling[70] (see Methods). As the likelihood function, which describes how well the model reproduces the rmsf values as a function of time during the T-quench simulations, we used a normal distribution. To disentangle the contribution of the size decrease to the decrease in rmsf values from other effects, we used the rmsf values obtained after rescaling the conformations (Fig. 2c, black lines).

The free model parameters optimized by the Bayes approach are $d$ and $c$ for model1, $\Delta G^{\ddagger}$, $\Delta x$, $\Delta G$ for model2, as well as $d$, $c$, $\Delta G^{\ddagger}$, $\Delta G$, and $\Delta x$ for model3. Using all model parameters for each quantile separately results in 10, 15, and 25 free parameters for model1, model2, and model3, respectively. The number of free parameters can be reduced by setting parameters to be the same for all quantiles. The different choices of which parameters are

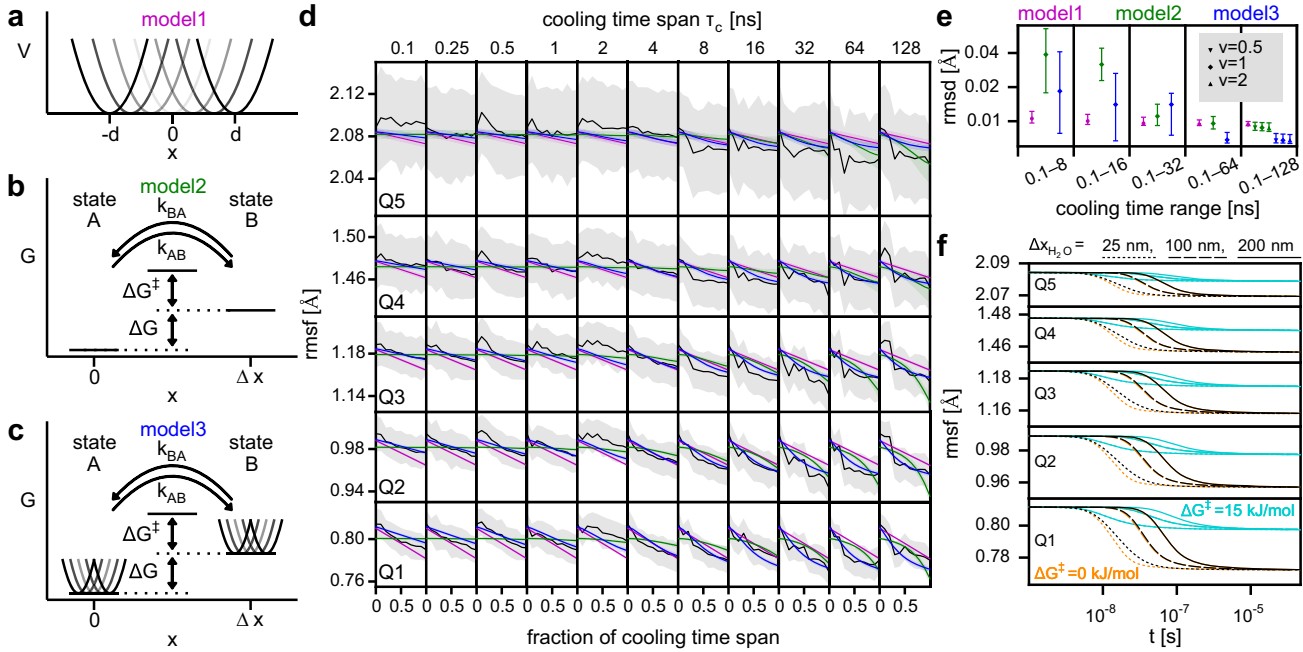

**Fig. 3 Models of cooling behavior.** Schematics of thermodynamic model1 with uniformly distributed harmonic potentials (**a**), kinetic two-state model2 (**b**), and combined model3 (**c**). **d** Comparison of the time dependence of the rmsf quantiles obtained from T-quench simulations after rescaling (black lines, compare Fig. 2c) with those obtained from the models (magenta, green, and blue lines). Standard deviations from simulations and the models are shown as gray and colored areas, respectively. **e** For each model, the root mean square deviation (rmsd) of the predicted rmsf values from the rmsf values obtained from the T-quench simulations (d, black lines) is shown. Rmsf values were obtained from 41 simulations for each cooling time span. To obtain model parameters, the T-quench rmsf values for different ranges of cooling time spans were used (0.1–8 ns, 0.1–16 ns, ... ). The symbols denote the mean values and the vertical lines correspond to the 95% confidence intervals obtained from the parameter distributions. **f** Rmsf decrease during plunge freezing as a function of time for the quantiles estimated from combining model3 with temperature drops (Fig. 1c) for different water-layer thicknesses ($\Delta x_{H_2O}$).

the same for all quantiles result in different variants of the model with different numbers of free parameters. To find the optimal model variant, we trained different variants on the rmsf values obtained for cooling time spans 0.1–64 ns (Fig. S5). The obtained distributions of the parameters were then used to calculate distributions of rmsf values as a function of cooling time for all cooling rates. The deviation of these model-derived rmsf values from the rmsf values used for training shows how well the model is able to reproduce the rmsf decrease in the T-quench simulations (Fig. S5, blue). To control for overfitting, we used cross-validation. To that aim, we first predicted rmsf values for a cooling time span of 128 ns. Next, we calculated their deviation from the corresponding values from the T-quench simulations (Fig. S5, red). For model1, model2, and model3, we chose the variant that shows the lowest cross-validation rmsd for further analysis. When the rmsds of two variants were not significantly different, we chose the variant with fewer free parameters. The chosen model1 variant has 6 free parameters (5 for $d$ and 1 for $c$), the chosen model2 variant has 11 free parameters (5 for $\Delta G^{\ddagger}$ and $\Delta x$, 1 for $\Delta G$), and the chosen model3 variant has 9 parameters (5 for $d$ and 1 for $c$, $\Delta G^{\ddagger}$, $\Delta G$, $\Delta x$). Each model was then trained on the rmsf values of all T-quench simulations (cooling time spans 0.1–128 ns) and the convergence of the Metropolis sampling was assessed by comparing probability densities of two independent calculations, which turned out to be very similar (Fig. S6).

For the three models, the distributions of the parameters were then used to calculate the distributions of rmsf values as a function of cooling time (Fig. 3d, magenta, green and blue). Since model1 does not take the cooling rate into account and, therefore, the rmsf only depends on the temperature, the rmsf as a function of the fraction of the cooling time span is the same for all cooling

time spans. This property results in an overestimation of the rmsf decrease with short cooling time spans (magenta line below black line) and an underestimation with longer cooling time spans (magenta line above black line).

The rmsf obtained from model2 is almost constant for short cooling time spans, because the temperature drops so rapidly that the barrier is not overcome and the probabilities of being in states A and B almost do not change. This behavior also results in an underestimation of the rmsf decrease for rapid cooling and an overestimation for slower cooling.

These results indicate that model1 and model2 do not fully capture the underlying physical processes and therefore do not predict the decrease in heterogeneity for longer cooling time spans very well. In contrast, the combined model3 captures both, the rmsf decrease during rapid cooling and the kinetic effect that the decrease is more pronounced during slower cooling.

All models either underestimate or overestimate the median values of the rmsf quantile Q5. However, the rmsf values obtained from the models lie very well inside the confidence intervals (Fig. 3d, gray area). Q5 quantifies the rmsf values of the atoms with the largest heterogeneity in the ensemble. We expect the underlying large conformational changes to be slower than conformational changes resulting in smaller rmsf values. Therefore, we expect the conformational changes underlying Q5 to be less equilibrated, which is supported by the larger confidence intervals.

To test how well the models predict rmsf decrease for a large range of cooling rates based on the information from fast cooling rates only, we trained the models on the rmsf values obtained from T-quench simulations with short cooling time spans, e.g., the range 0.1–8 ns. From the obtained parameter distributions,

the rmsf values for all cooling time spans (0.1–128 ns) were predicted and compared to the rmsf values obtained from the corresponding T-quench simulations (Fig. 3e). For the thermodynamic model1, the deviation of the predicted values from the T-quench simulation values did not decrease markedly when rmsf values from longer cooling time spans were used to train the model. The deviations are similar, because the parameter distributions obtained from the different training sets were similar, suggesting that the contribution to the rmsf decrease described by this model, namely the equilibration in local harmonic potentials, is indeed cooling-rate independent. In contrast, for model2 and model3, the deviation decreased markedly when more simulation data was used for training. Model3 showed the lowest deviations of all models when cooling time spans ranges of 0.1–64 ns and 0.1–128 ns were used. The choice of the temperature exponent $v$ does not significantly affect the agreement (Fig. 3e, last column, green symbols) and in the following we will use $v = 1$. Interestingly, for model3, although the deviations are larger than for model2 when only short cooling time spans were used for training, model3 predicts large confidence intervals which contain small rmsd values. These results indicate that model3 accurately predicts the rmsf decrease and that more data for training primarily results in less uncertainty of the prediction.

One might argue that a combined model will always be better than the individual two models. However, the fact that the optimal model3 variant has fewer free parameters (9) than the optimal model2 variant (11) indicates that it is the underlying physics of the model and not the number of parameters that result in an improved prediction capability of model3. Taken together, our Bayes approach suggests that a combination of static structural heterogeneity and cooling-rate dependent barrier crossing best describes the narrowing of the structure ensembles observed in all T-quench simulations combined. Therefore, we will subsequently use model3 to describe the effects of cooling on structure ensembles.

**Effects of plunge freezing on the ensemble.** The main aim of this work is to quantify the effect of plunge-freezing on the heterogeneity of the frozen structural ensemble that is used for the cryo-EM measurements. To quantify the effect for realistic cooling protocols, we combined the estimated temperature drop during plunge-freezing (Fig. 1c) with the kinetic model that best describes the rmsf decrease during the T-quench simulations (Fig. 3c). Figure 3f shows the rmsf decrease as a function of the cooling time calculated using the kinetic model with the obtained model parameters (Fig. S6c) and the temperature drops for three different water-layer thicknesses (Fig. 1c, 25 nm, 100 nm, and 200 nm). As expected, thinner water layers which lead to more rapid temperature drops result in more rapid narrowing of the structural ensembles.

However, our model suggests that the final rmsf values are rather independent of the water-layer thickness within the range that we probed, because the barriers are small enough to be overcome during the cooling process such that the final rmsf values do not depend on the barrier height. The final rmsf values are decreased to 95.2% (standard deviation: 0.4%) of their value before cooling for Q1 and to 99.30% (0.06%) for Q5. To put these results into the context of experimental results, we use B-factors, sometimes called temperature factors, atomic displacement parameters or Debye-Waller factors. The B-factor is a measure of the displacement of atoms around their mean position that can be obtained from x-ray and neutron scattering data and directly relates to the rsmf, $B = 8\pi/3 \cdot \text{rmsf}^2$ (ref. [71]). The calculated decrease in rmsf corresponds to a decrease in the B-factor of 0.52 Å² (standard deviation 0.04 Å²) which is similar for each quantile.

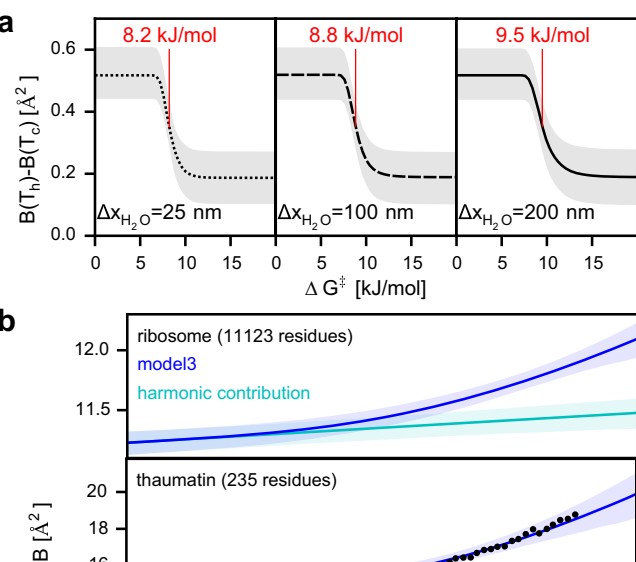

**Fig. 4 B-factor dependency on barrier height and temperature.** **a** Difference between B-factors at the temperatures before cooling ($T_h$) and after cooling ($T_c$) calculated from model3 as a function of barrier height $\Delta G^{\ddagger}$ for different water-layer thicknesses ($\Delta x_{H_2O}$). The $\Delta G^{\ddagger}$ values at which the B-factor is reduced halfway between the reduction at 0 kJ/mol and 20 kJ/mol is indicated in red. Lines denote expected values and gray areas correspond to the 95% confidence intervals (obtained from bootstrapping the parameter distributions). **b** B-factors as a function of temperature calculated from model3 under equilibrium conditions (upper panel; expected values: blue lines, 95% confidence interval: light blue area). The B-factors of the harmonic potentials are shown in cyan. B-factors obtained from X-ray crystallography at different temperatures (black dots) for proteins thaumatin[60] (middle panel) and ribonuclease-A[40] (lower panel). B-factors calculated from model3 applied to the experimental B-factors (blue and cyan).

In the T-quench simulations, only barriers are overcome during the atomistic cooling simulations contribute to the rmsf decrease and are therefore described by the model. Accordingly, a high free-energy barrier $\Delta G^{\ddagger}$ of 15 kJ/mol leads to a smaller decrease (Fig. 3f, cyan lines). To probe how the B-factor decrease depends on $\Delta G^{\ddagger}$ and the water-layer thickness, we applied model3 to the estimated plunge-freezing temperature drops with the parameters obtained from our Bayes approach (Fig. S6c) except for $\Delta G^{\ddagger}$ which was set to values between 0 and 20 kJ/mol. The B-factor decrease during cooling as a function of the barrier height $\Delta G^{\ddagger}$ shows a sharp transition between small and large barrier heights (Fig. 4a). The $\Delta G^{\ddagger}$ value where the transition occurs is highlighted in red for the very different cooling rates obtained for different water-layer thicknesses and is shifted to slightly larger values for thicker water layers. This shift to larger barrier heights is expected, because with thicker water layers the temperature drop is slower which results in more time to overcome larger barriers.

In summary, from our analysis, we expect that in addition to the equilibration of atoms in local harmonic potentials, conformational

changes can contribute to the decrease in structural heterogeneity during plunge-freezing. For conformational changes with a free-energy barrier below a certain threshold, the barrier can be overcome during cooling and the conformation with lower free energy is predominately occupied, therefore decreasing the structural heterogeneity. A thicker water layer results in slower cooling such that higher barriers can be overcome leading to a more homogeneous ensemble.

**Kinetic cooling model shows glass transition**. The dynamic behavior of proteins undergoes a glass transition at temperatures of around 200 K[61]. Above the transition temperature which has been found to be similar for different proteins[34,36,40,42,43,46–51,53,61,62], the slope of the average B-factor as a function of temperature is markedly higher than below the transition temperature. This observation provides an opportunity to test if our model3 (Fig. 3c) reproduces this ubiquitous behavior. To that aim, we calculated the B-factors from the model with the parameters obtained from the Bayes approach (Fig. S6c) and compared them to B-factors from x-ray crystallography. Since the temperature drop for crystals is much slower than for thin water layers[55], we assumed the two states to be in equilibrium at any temperature, i.e., the probability of being in state A or B is determined only by the free-energy difference $\Delta G$ and the temperature $T$. The B-factors as a function of temperature calculated from the model for the Q3 quantile are shown in Fig. 4b (upper panel, blue line) and for the other quantiles in Fig. S7a. Additionally, we show the contribution of the harmonic potentials (cyan line) by setting parameters $\Delta G$ and $\Delta x$ to 0 kJ/mol and 0 Å, respectively. At low temperatures, the probability of being in the state with higher free energy (state B) is negligible. Therefore the increase of the B-factors is governed by the harmonic potentials as indicated by the overlap of the B-factors from the harmonic contribution (cyan) and from the full model (blue). Around a temperature of 200 K, our model predicts a markedly steeper increase of the B-factors which is in agreement with the experimentally observed protein glass transition temperatures and results from the non-harmonic contributions. For higher temperatures, the occupancy of conformational states with higher free energies is increased (state B in our model) giving rise to a steeper B-factor increase.

The temperature dependency of the B-factors predicted by our model trained on the ribosome T-quench simulations is similar to that of the B-factors obtained from X-ray crystallography of proteins thaumatin[60] and ribonuclease-A[40] (Fig. 4b, black dots). However, the B-factors at high temperatures as well as the B-factor difference between high and low temperatures are different for the three cases as expected for macromolecules with different sizes and structures. In addition, the finding that these values differ for the two protein crystal structures suggest that the differences for the three cases are not a result of the different methods used to obtain these values, but rather that they are specific for the chosen macromolecules. To further compare the B-factor dependency of the three macromolecules, we trained the equilibrium model3 on the x-ray B-factors (Fig. 4b). Interestingly, the probability densities of the model parameters $\Delta x$, $\Delta G$, and $c$ for model3 trained on the T-quench simulations and on the two x-ray structures overlap (Figs. S6c and S7b). These overlaps show that the same values of these parameters can describe the B-factor dependencies for these very different macromolecules. Only the probability densities of the parameter $d$, which determines the absolute B-factor at low temperatures, do not overlap. Taken together, these results further suggest that our simple model captures the effects of cooling on the B-factors of a broad range of biomolecules.

## Discussion

In addition to achieving high resolution, the recent revolution in cryo-EM facilitated the determination of ensembles of structures in different states[18–21,72]. These structural ensembles are obtained from EM images of samples that were rapidly cooled down to cryogenic temperatures by plunge-freezing[13]. Generally, biomolecules thermodynamically access more conformations at room temperature than at the cryogenic temperature and rates of conformational changes are determined by free-energy barriers and temperatures[22]. For very rapid cooling, low temperatures that prevent barrier crossing are quickly reached and the conformations prior to cooling are expected to be kinetically trapped. In contrast, during very slow cooling, most barriers are expected to be overcome and low free-energy conformations are predominately occupied resulting in an ensemble that is more homogeneous than the room-temperature ensemble prior to cooling. However, to what extent the rapid cooling perturbs the ensembles is currently unknown.

Here, we quantify the effects of plunge-freezing on structural ensembles of biomolecules. Our approach is divided into three layers of modeling: (1) a continuum model to estimate the temperature drop during plunge-freezing with different experimentally observed water-layer thicknesses, (2) calculation of the response of a macromolecular system to different cooling rates (T-quench MD simulations of ribosome complex), and (3) several kinetic models trained against the MD simulation data and then applied to the temperature drops estimated by model layer (1).

Our results suggest that three effects contribute to the narrowing of the structural ensembles, resulting in smaller B-factors: first, the thermal contraction of the biomolecules, second, the equilibration within local potential wells, and third, the equilibration into lower free-energy conformations by overcoming the barriers that separate the conformations. The first two effects appear to be largely cooling-rate independent in contrast to the third effect where the cooling rate determines the barrier heights that are overcome during cooling. Our results suggest that this barrier-height threshold is between 8 and 10 kJ/mol for realistic cooling rates are overcome and that the B-factor is reduced by 0.5 Å² in the ensemble imaged by cryo-EM compared to the ambient-temperature ensemble. In summary, our combined approach provides quantitative data on how much, for given cooling speed, the ambient temperature ensemble narrows, as well as a quantitative relationship between the heights of the barriers that are overcome during cooling and the temperature drop during cooling.

In our continuum model calculation of the water-layer temperature drop during plunge-freezing, we assumed that the water layer comes into contact with the liquid ethane instantaneously and the heat exchange is via conduction only. Our calculations yielded cooling rates consistent with cooling rates required for vitrification ($>10^6$ K/s)[13]. By high-speed photographic imaging it was observed that the grid, when plunged into the liquid ethane, initially displaces the ethane and only a fraction of the circumference of the grid comes into direct contact with the ethane immediately[73]. From this observation it was concluded that vitrification might instead be achieved by heat conduction through the grid. This mechanism might result in cooling rates different from our estimated rates but also not slower than the vitrification limit of $10^6$ K/s. We tested this possibility with our continuum model applied to a water layer with a width of 3 mm (corresponding to the cryo-EM grid diameter) with a liquid-ethane layer on one side and air on the other side and found that after one second, temperatures below 200 K are only reached within a distance of 0.2 mm from the circumference. Within this time the grid would be fully submerged in liquid ethane, such that, except for a small region close the circumference of the grid,

the heat conduction through the grid can not markedly contribute to the cooling. This result suggests that our assumption that the cooling is dominated by direct contact between the water-layer and the ethane is likely to be valid.

Above the liquid cryogen, a cold gas layer with a thickness of several mm was observed[74,75]. To what extent the cryo-EM samples are already cooled when they move through the gas layer during plunging is not clear. The slower temperature drop due to precooling by the cold gas layer, would allow biomolecules to overcome higher barriers resulting in more homogeneous ensembles. We are not aware of any direct measurement of the temperature drop in thin cryo-EM samples during plunge-freezing. Should new data become available, any measured cooling rate could be included within our kinetic model in a straightforward way, thus refining our estimate of how plunge-freezing affects structural ensembles.

To disentangle the contributions of thermal contraction, equilibration within local potential wells, and equilibration into lower free-energy conformations to the narrowing of structural ensembles, we performed T-quench MD simulations with different cooling rates and subsequently trained and cross-validated thermodynamic and kinetic models. For the T-quench simulations, we used cooling rates that are realistic for the onset of cooling, where the temperature is still high enough to enable conformational changes, and therefore the expected effect on the structural ensemble is largest. Combined with the kinetic models, this approach allowed us to overcome the problem of simulating the slow cooling 'tails' at low temperatures, which would be computationally intractable.

Our approach permitted to separate the narrowing of structural ensembles from simple thermal expansion. Indeed, the thermal expansion coefficient obtained from the T-quench simulations of a ribosome complex seems to be largely independent of the cooling rate and agrees with coefficients obtained from x-ray crystallography and MD simulations of proteins[40,54]. This observation now allows one to quantify the contribution of thermal contraction to B-factors and to determine the size of biomolecules at room temperature from the cryo-EM structures of different types of biomolecules.

To quantify if and how much the two effects, equilibration within local potential wells and the equilibration into lower free-energy conformations by overcoming the barriers, contribute to the narrowing of structural ensembles, we used the results of the T-quench simulations to train and validate three different models of the cooling process. We found that the model which combines both effects matches and predicts the T-quench simulations significantly better than the two models which considers only one of the two effects. The combined model allowed us to disentangle the contributions of the two effects and to address the question of how different temperature drops and different barrier heights affect the B-factor decrease (Figs. 3f and 4a). The observation that our model predicts a glass transition temperature within the range found with very different methods for other biomolecules[34,36,40,42,43,46–51,53,61,62] provides an independent confirmation of our approach.

We note that during cold denaturation a transient increase of the accessible number of conformations can occur during cooling[76], which, however, is not relevant to cryo-EM studies, because during plunge-freezing the biomolecule typically reaches the glass-transition temperature on timescales much shorter than those expected for unfolding. In our simulations, we have not observed unfolding events either, which provides additional support for this notion.

We hypothesize that the dependence of the trapped conformation on the temperature prior to cooling observed in some cryo-EM experiments[24–26] might be due to temperature dependent free-energy differences between conformational states.

Although our MD simulations fully capture how free energies change with temperature and even the associated non-equilibrium effects, we here started all simulations from an ensemble at one (ambient) temperature, such that this effect cannot be explored. For this reason, we have also not included temperature dependent free-energy differences between conformational states within our simple kinetic models. One might test this hypothesis by generating ensembles at several temperatures, then starting T-quench simulations from these ensembles, and quantifying the occupancy of conformational states during cooling.

In cryo-EM experiments, the B-factors result from several effects. Apart from thermal motion of the atoms, kinetic trapping in different conformations during cooling and the thermal contraction, also detector resolution, inaccurate sorting and alignment errors contribute to the obtained B-factors. Obtaining cryo-EM reconstructions from biomolecules that were cooled at different cooling rates would allow to quantify which contribution arises from the kinetic trapping and to determine the distribution of free-energy barrier heights. Further, cryo-EM reconstructions from biomolecules that were cooled down in the same way, but imaged at different temperatures (below the glass transition temperature), would allow to quantify the contribution of thermal motion. Effects that are independent of both, the cooling rate and the temperature of imaging, could then be attributed to the other effects. A promising way to alter the cooling rate in a controlled manner is provided by microfluidic cryofixation[77]. The geometry of the microfluidic channel determines the cooling rate such that different cooling rates should be achievable. Finally, we would expect cooling rates to depend on the position in the ice layer, e.g., depending on the distance to the surface of the sample layer. Reconstructions from different positions could further help to disentangle the effects.

The found relationship between the heights of the barriers that are overcome during cooling and the temperature drop during cooling bears several implications for ways to optimize and interpret cryo-EM results. To obtain high-resolution structures, where a homogeneous ensemble of structures is required, one would start cooling from a low temperature ensemble (above the melting point of water) and then cool down as slow as the vitrification limit allows. In contrast, to achieve the best representation of a room-temperature ensemble, one would cool down as rapidly as possible to avoid relaxation into nearby free-energy minima. Our findings suggest that distinct conformations resolved in cryo-EM experiment are separated by barriers larger than ~10 kJ/mol with current cooling protocols. Our observed reduction in B-factors can be used to rescale measured B-factors and thus to obtain better estimates for room-temperature ensembles. This simple scaling procedure rests of course on the assumption — supported by the glass transition studies mentioned above — that the scaling factor is similar for different proteins and protein complexes. Application of our combined approach, including the atomistic simulations, to any molecular system of interest also allows one to drop or test this assumption. We are confident that our gained detailed understanding of how structural ensembles are affected by the cooling rate will inspire experiments where both the temperature prior to cooling and the cooling rate are varied. Then, analysis of the observed occupancies of different conformations based on our framework has the potential to quantify not only the free-energy differences between conformations at room temperature but also the heights of separating barriers.

## Methods

**Cooling rate estimation**. To estimate the temperature drop of a film of water plunged into liquid ethane, we numerically solved the heat equation for a layer of

water starting at $T = 277.15$ K between two layers of ethane starting at $T = 90$ K. The temperature at the outer boundaries was fixed to $T_b = 90$ K. To characterize the effect of the thicknesses of the layers, we used water-layer thicknesses of 25 nm, 100 nm, and 200 nm as well as ethane layer thicknesses of 100 nm, 200 nm, 400 nm, 800 nm, 1.6 μm, and 3.2 μm. For ethane, we used the thermal diffusivity $\alpha = 0.173$ mm²s⁻¹ calculated from the thermal conductivity, the specific heat capacity, and the density at a temperature of $T = 90.35$ K[78]. For water, we used $\alpha = 0.133$ mm²s⁻¹ calculated from the thermal conductivity[79], the specific heat capacity[80], and the density[79] at a temperature of $T = 273.15$ K. The heat equation is $\partial_t T(x, t) = \partial_x [\alpha(x) \partial_x T(x, t)]$, where $x$ is the 1-d coordinate and $\alpha(x)$ is the thermal diffusivity at position $x$. This equation was solved numerically and more details can be found in the Supplementary Methods (Numerical solution of the heat equation for a water layer).

**Structural ensemble before cooling.** To generate an ensemble of structures of the ribosome · EF-Tu · kirromycin complex at the temperature before cooling (277.15 K), we used all-atom explicit-solvent molecular dynamics (MD) simulations starting from a high-resolution cryo-EM structure[30]. We used the same system setup and pre-equilibration protocol as described earlier for MD simulations of the same system[69], except that the temperature was set to 277.15 K and that the production simulation was extended to 3.5 μs. The simulations were carried out using GROMACS 5.1[81], with the amber99sb force field[82], the SPC/E water model[83], and K⁺Cl⁻ ion parameters by Joung et al.[84]. Bond lengths were constrained using the LINCS algorithm[85]. Virtual site constraints for hydrogens[86] allowed a 4-fs integration step.

**Convergence with regards to structural heterogeneity.** We used the root mean square fluctuation (rmsf) of atomic positions calculated for an ensemble of structures as a measure for the structural heterogeneity of the ensemble. To assess convergence of the 277.15 K simulation with regards to this observable, we first extracted structures from the trajectory at intervals of 50 ns. The extracted structures were aligned via rigid-body fitting using the Cα-atoms of the amino acids and the P-atoms of the nucleotides. Then we grouped all 41 structures between 0 μs and 2 μs into one ensemble (Fig. S1a, top) and calculated the rmsf for all atoms resolved in the cryo-EM structure[30] (Fig. S1b). This calculation was repeated for the ensembles consisting of structures between 0.1 μs and 2.1 μs, between 0.2 μs and 2.2 μs, up to between 1.5 μs and 3.5 μs.

**T-quench simulations.** To estimate the effect of different cooling rates on the ensemble, we carried out MD simulations with different cooling time spans $\tau_c \in$ [0.1ns, 0.25ns, 0.5ns, 1ns, 2ns, 4ns, 8ns, 16ns, 32ns, 64ns, 128ns]. These T-quench simulations were started from 41 structures extracted from the 277.15 K trajectory at 1000 ns, 1050 ns, ... , 3000 ns (Fig. 2a). For each cooling time span $\tau_c$, 41 simulations of length $\tau_c$ were carried out with the temperature linearly decreasing from 277.15 K to 77 K. The temperatures of solute and solvent were controlled independently using velocity rescaling[87] with a coupling constant of $\tau_T = 0.1$ ps. The pressure was coupled to a Parrinello-Rahman barostat[88] ($\tau_p = 1$ ps).

**Structural heterogeneity during T-quench simulations.** To quantify the effects of cooling on the structural heterogeneity of the simulated ensemble, we calculated rmsf values for each atom in the ensembles before, during, and at the end of cooling. To that aim, we first extracted the coordinates of all atoms of the ribosome complex which were resolved in the cryo-EM structure from the frames of the trajectories obtained from the T-quench simulations. For each cooling time span $\tau_c$, the 41 T-quench trajectories were analyzed as an approximation to the non-equilibrium ensemble. From each trajectory, the coordinates were extracted at 11 time points $t_j$ with $t_j \in$ [0ns, 0.1$\tau_c$, 0.2$\tau_c$, ..., $\tau_c$]. Subsequently, for each cooling time span $\tau_c$ and each time point $t_j$, we then grouped the 41 structures from the 41 simulations in an ensemble. Next, for each ensemble the structures were aligned via rigid-body fitting using the Cα- and the P-atoms. Then for each atom the rmsf was calculated (Fig. 2b).

For each cooling time span $\tau_c$, the rmsf values for all $N$ atoms are collected in the matrix

$$
R_{\tau_c} = \begin{bmatrix}
r_{1,\tau_c}(0\,\text{ns}) & r_{1,\tau_c}(0.1\tau_c) & r_{1,\tau_c}(0.2\tau_c) & \cdots & r_{1,\tau_c}(\tau_c) \\
r_{2,\tau_c}(0\,\text{ns}) & r_{2,\tau_c}(0.1\tau_c) & r_{2,\tau_c}(0.2\tau_c) & \cdots & r_{2,\tau_c}(\tau_c) \\
\cdots & \cdots & \cdots & \cdots & \cdots \\
r_{N,\tau_c}(0\,\text{ns}) & r_{N,\tau_c}(0.1\tau_c) & r_{N,\tau_c}(0.2\tau_c) & \cdots & r_{N,\tau_c}(\tau_c)
\end{bmatrix}.
$$

Each row of $R_{\tau_c}$ contains all rmsf values of a certain time point $t_j$ for a certain cooling time span $\tau_c$. To describe the distributions of rmsf values and to quantify changes during cooling, we calculated the 6-quantiles, i.e., the 5 rmsf values that divide the set of atoms into six equally sized subsets of atoms. The quantiles are

collected in the matrix

$$
Q_{\tau_c} = \begin{bmatrix}
Q_{1,\tau_c}(0\,\text{ns}) & Q_{1,\tau_c}(0.1\tau_c) & Q_{1,\tau_c}(0.2\tau_c) & \cdots & Q_{1,\tau_c}(\tau_c) \\
Q_{2,\tau_c}(0\,\text{ns}) & Q_{2,\tau_c}(0.1\tau_c) & Q_{2,\tau_c}(0.2\tau_c) & \cdots & Q_{2,\tau_c}(\tau_c) \\
\cdots & \cdots & \cdots & \cdots & \cdots \\
Q_{5,\tau_c}(0\,\text{ns}) & Q_{N,\tau_c}(0.1\tau_c) & Q_{N,\tau_c}(0.2\tau_c) & \cdots & Q_{N,\tau_c}(\tau_c)
\end{bmatrix},
$$

where $Q_{i,\tau_c}(t_j)$ is the $i$-th quantile of the distribution of rmsf values at time point $t_j$ for cooling time span $\tau_c$. To estimate the uncertainty of the 6-quantiles, we carried out bootstrapping of the structures in each ensemble resulting in the matrices $S_{\tau_c}$ containing the standard deviations $\sigma_{i,\tau_c}(t_j)$ of each of the $Q_{i,\tau_c}(t_j)$ value.

**Size decrease during cooling.** We observed a decrease in the size of the ribosome over the course of the T-quench simulations. To quantify the decrease, we introduced a scaling factor $s$ that is multiplied to all the extracted coordinates after placing the center of geometry in the origin. For each $t_j > 0$ ns and each $\tau_c$, we calculated the scaling factor $s$ that minimizes the average root mean square deviation (rmsd) of all 41 structures from to their respective starting structure ($t_j = 0$ ns). Figure S3 shows the scaling factor $s$ for all $t_j$ and $\tau_c$. To disentangle the effect of size decrease from the reduced structural heterogeneity, we then calculated the rmsf quantiles $Q_{\tau_c}$ and standard deviations $S_{\tau_c}$ from the scaled structures and used those in the following analysis (Fig. 2c).

**Thermodynamic and kinetic models of the cooling process.** To check if the decrease in conformational heterogeneity during cooling depends on the rate of cooling, we considered one thermodynamic and two kinetic models of the cooling process. The first model describes the rmsf of atoms in uniformly distributed harmonic potentials in equilibrium and is therefore independent of the cooling rate. The second model describes the rmsf of atoms by jumps between two states at different free energies separated by a free-energy barrier. The third model is a combination of the first two models.

*Model1: uniformly distributed harmonic potentials.* The probability density $p_h(a, b, x)$ for an atom in harmonic potentials whose mean values are uniformly distributed between $a$ and $b$ is given by

$$
p_h(a, b, x) = \int_a^b \frac{1}{b-a} \cdot \frac{1}{\sigma \sqrt{2\pi}} \exp\left(-\frac{(x-\mu)^2}{2\sigma^2}\right) d\mu \tag{1}
$$

$$
= \frac{1}{2(a-b)} \left[ \text{erf}\left(\frac{x-b}{\sigma\sqrt{2}}\right) - \text{erf}\left(\frac{x-a}{\sigma\sqrt{2}}\right) \right] \tag{2}
$$

with $\sigma = \sqrt{\frac{k_B T}{c}}$, where $k_B$ is the Boltzmann constant, and $c$ is the force constant of the harmonic potentials. The rmsf of the atom is then given by

$$
\text{rmsf}_h(a, b, \bar{x}) = \sqrt{\int_{-\infty}^{\infty} (x - \bar{x})^2 p_h(a, b, x) dx} \tag{3}
$$

$$
= \sqrt{\sigma^2 + \frac{a^2 + ab + b^2}{3} - (a+b)\bar{x} + \bar{x}^2}, \tag{4}
$$

where $\bar{x}$ is the expectation value of $x$. For model1, we set $a$ to $-d$ and $b$ to $+d$ (compare Fig. 3a). Therefore, $\bar{x} = 0$ and

$$
\text{rmsf}_{\text{model1}} = \text{rmsf}_h(-d, d, 0) = \sqrt{\frac{k_B T}{c} + \frac{d^2}{3}}. \tag{5}
$$

*Model2: kinetic two-state model.* In the two-state model, each atom can visit two states $A$ and $B$ that are separated by a distance $\Delta x$ along a 1-d coordinate $x$ (Fig. 3b). The free energy of state $B$ is chosen to be larger than that of state $A$ by $\Delta G$. Additionally, the two states are separated by a free-energy barrier $\Delta G^\ddagger$.

Under the assumption that the two states are in equilibrium at temperature $T_h$ before cooling, the probability of being in states $A$ and $B$ is $P_A = 1/(1 + \exp(-\Delta G/(k_b T_h)))$ and $P_B = 1/(1 + \exp(\Delta G/(k_b T_h)))$, respectively. For the non-equilibrium time evolution during cooling, the rates between the two states contribute to the probabilities and are given by a modified Arrhenius equation,

$$
k_{AB}(t) = \kappa \left(\frac{T(t)}{T_h}\right)^\nu \exp\left(-\frac{\Delta G + \Delta G^\ddagger}{k_b T(t)}\right),
$$

$$
k_{BA}(t) = \kappa \left(\frac{T(t)}{T_h}\right)^\nu \exp\left(-\frac{\Delta G^\ddagger}{k_b T(t)}\right),
$$

where $\kappa \left(\frac{T(t)}{T_h}\right)^\nu$ is the pre-exponential factor with temperature $T$ depending on time $t$. $\nu$ is an exponent of the temperature and $\kappa$ a scaling factor. In our T-quench simulations the temperature was linearly decreased from $T_h$ at $t = 0$ to $T_c$ at $t = \tau_c$. Hence, $T(t) = T_h - \alpha t$ with $\alpha = (T_h - T_c)/\tau_c$. The Master equation for the

probability of being in state $A$ as a function of time $P_A(t)$ is then

$$\frac{dP_A(t)}{dt} = k_{BA}(t)\big(1 - P_A(t)\big) - k_{AB}(t)P_A(t).$$

Inserting the rates $k_{AB}(t)$ and $k_{BA}(t)$ results in the differential equation

$$\frac{dP_A(t)}{dt} = \kappa \left(\frac{T(t)}{T_h}\right)^{\nu} \left[ \exp\left(-\frac{\Delta G^{\ddagger}}{k_b(T_h - \alpha t)}\right)\big(1 - P_A(t)\big) - \exp\left(-\frac{\Delta G + \Delta G^{\ddagger}}{k_b(T_h - \alpha t)}\right)P_A(t)\right]. \tag{6}$$

For given values of $\kappa$, $\Delta G$, $\Delta G^{\ddagger}$, and $\nu$, the differential equation for $P_A(t)$ was solved numerically using the function odeint of SciPy[89]. The rmsf of an atom described by the model as a function of time can then be calculated by

$$\mathrm{rmsf}_{\mathrm{model2}}(t) = \Delta x \sqrt{P_A(t)(1 - P_A(t))}. \tag{7}$$

*Model3: kinetic two-state model with uniformly distributed harmonic potentials.* To capture possible kinetics during cooling and to include the uniformly distributed harmonic potentials, we combined the two models described above (Fig. 3c). Here, as in the model2, $P_A(t)$ and $P_B(t) = 1 - P_A(t)$ are the probabilities of the atom being in states $A$ and $B$ obtained from numerically solving Eq. (6). Extending the two-state model, the probability in states $A$ and $B$ is governed by harmonic potentials uniformly distributed from $-d$ to $d$ and from $\Delta x - d$ to $\Delta x + d$, respectively. The probability density as a function of time $t$ is then given by $p(x, t) = P_A(t)p_h(-d, d, x) + P_B(t)p_h(\Delta x - d, \Delta x + d, x)$, where $p_h$ is given by Eq. (2). The probability density results in a mean position of the atom $\bar{x}(t) = P_B(t)\Delta x$. The rmsf is then obtained from

$$\mathrm{rmsf}_{\mathrm{model3}}(t) = \sqrt{\sqrt{\int_{\infty}^{\infty}(x - \bar{x}(t))^2 p(x, t)dx}} = \sqrt{P_A(t)r_A(t) + P_B(t)r_B(t)} \tag{8}$$

with

$$r_A(t) = \int_{\infty}^{\infty}(x - \bar{x}(t))^2 p_h(-d, d, \bar{x}(t))dx$$
$$= [\mathrm{rmsf}_h(-d, d, \bar{x}(t))]^2$$

and

$$r_B(t) = \int_{\infty}^{\infty}(x - \bar{x}(t))^2 p_h(\Delta x - d, \Delta x + d, \bar{x}(t))dx$$
$$= [\mathrm{rmsf}_h(\Delta x - d, \Delta x + d, \bar{x}(t))]^2,$$

where $\mathrm{rmsf}_h$ is given by Eq. (4).

**Finding the parameters of the models.** The three models of the cooling process have different sets of parameters $c$ and $d$ for model1, $\Delta x$, $\Delta G$, and $\Delta G^{\ddagger}$ for model2, as well as all five parameters for the combined model3. We considered different variants of the models where each parameter can either be the same value for all five quantiles or one value for each quantile. To obtain probability densities of the parameters, we used a Bayes approach; the details can be found in the Supplementary Methods (Metropolis sampling with Bayesian inference).

To obtain the optimal combination of numbers of parameters, we trained models with different numbers of parameters (Fig. S5) and compared how well they reproduced the rmsf values they were trained on and how well they predict rmsf values not included in the training; the details can be found in the Supplementary Methods (Finding optimal model variants).

**Decrease of structural heterogeneity during plunge-freezing.** To estimate the decrease in structural heterogeneity during the plunge-freezing, we applied model3 with the obtained parameters (Fig. S6c) to the temperature drops estimated for different water-layer thicknesses (Fig. 1c). After omitting the first 20% of steps, from 1000 randomly chosen Metropolis steps, parameters were extracted and 1000 sets of rmsf curves were calculated from the model. To investigate the influence of the barrier height $\Delta G^{\ddagger}$, from the same parameters rmsf curves were calculated after setting $\Delta G^{\ddagger}$ to 0 kJ/mol or to 15 kJ/mol. The median and 95% confidence intervals of the rmsf curves are shown in Fig. 3f.

**B-factors as a function of temperature and barrier height.** To compare the results of model3 to B-factors obtained from experiments, e.g., x-ray crystallography at different temperatures[40], we calculated B-factors from the model at different temperatures with the parameters from 1000 randomly chosen Metropolis steps (Fig. 4a, blue lines). The contribution of the harmonic potentials to the B-factors was calculated by evaluating $\mathrm{rmsf}_{\mathrm{model1}}$ (Eq. (5)) with the chosen $c$ and $d$ parameters (Fig. S7a, cyan lines). Next, we trained model3 using the B-factor values for ribonuclease-A obtained from x-ray crystallography[40] to obtain probability densities for parameters $\Delta x$, $\Delta G$, $c$, and $d$ (Fig. S7b). Since cooling of crystals is much slower than for the thin water layers used in cryo-EM[55], we set the $\Delta G^{\ddagger}$ parameter to 0 kJ/mol such that there is no kinetic contribution. We calculated B-factor values as a function of temperature (Fig. 4b) as described above (Fig. 4a).

To study the effect of the barrier height $\Delta G^{\ddagger}$ on the reduction of B-factors, we applied model3 to the temperature drops (Fig. 1c) with 1000 sets of parameters chosen as described above. Here, however, we set the barrier height $\Delta G^{\ddagger}$ to values

between 0 kJ/mol and 20 kJ/mol. The median and confidence intervals of the differences between the B-factors before cooling ($T_h$) and after cooling ($T_c$) are shown as a function of $\Delta G^{\ddagger}$ (Fig. 4c).

**Reporting summary.** Further information on research design is available in the Nature Research Reporting Summary linked to this article.

## Data availability
Ribosome-complex structures of the ensembles before and after cooling for all cooling time spans, the rmsf quantiles obtained from MD simulations, and the temperature drop estimates are available on Zenodo (https://doi.org/10.5281/zenodo.5948727). Additional data is available from the corresponding author upon request.

## Code availability
The code to train and and analyze kinetic model3 is available on Zenodo (https://doi.org/10.5281/zenodo.5948727). Additional code is available from the corresponding author upon request.

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

## Acknowledgements

The work was funded by the Deutsche Forschungsgemeinschaft (DFG, German Research Foundation) under Germany's Excellence Strategy - EXC 2067/1- 390729940 (L.V.B and H.G.). Computer time has been provided by the Max Planck Computing and Data Facility and the Leibniz Rechenzentrum. We thank Kresten Lindorff-Larsen, Iris Young, and James Fraser for their valuable suggestions.

## Author contributions

L.V.B. performed and analyzed the continuum model calculations, the MD simulations and the Metropolis sampling of the kinetic models. L.V.B. and H.G. conceived the project, interpreted the results, and wrote the paper.

## Funding

## Competing interests

The authors declare no competing interests.
