## [Peer Review File · Nature Communications]

Effects of cryo-EM cooling on structural ensemblesReviewers' Comments:

Reviewer #1:

Remarks to the Author:

The manuscript by Bock & Grubmüller describes a detailed, multi-pronged computational study of the complex and important effects of cooling during sample preparation for cryo-EM. The paper is generally easy to read, appears technically sound and provides relatively clear results that will be of interest both to theoreticians and practitioners of cryo-EM.

Over the last 10 years cryo-EM has delivered increasingly high-resolution structures that in some cases now rival those of e.g. X-ray crystallography. In addition to examining the structures of macromolecules, cryo-EM may also provide more detailed insights into their energy landscapes because it in principle is a single-molecule technique that enables the visualization of the conformational distribution of molecules.

These advances leave two questions that have been difficult to answer. First, to what extent does the "average" structure under cryogenic conditions reflect the ambient temperature "average" structure [realizing that the term average here is somewhat misleading, the authors will understand what is meant]. Second, to what extent does the distribution of conformations (conformational ensemble) present in the cryo-EM sample reflect the distribution at ambient temperatures [leaving aside the technical difficulties of determining structural models of these ensembles from experiments]. While the first question can to a certain extent be answered by comparing structures solved at cryo-conditions with those at ambient temperature (by crystallography), the second question lies at the heart of the utility (and large potential) for cryo-EM to study conformational ensembles.

This study provides welcomed data in an area that has been lacking detailed and quantitative modelling, and where experiments are difficult. The results are promising in the sense that they support the idea that cryo-EM can to a large extent capture conformational ensembles at ambient temperatures. Importantly, the study provides a framework to think about these problems in a more quantitative manner that will hopefully spur additional experiments and analyses.

Specific comments:

Major

p. 4:

I must admit that I found the RMSF-based analysis somewhat difficult to follow in places. First, just to be sure could the authors confirm that in each case the RMSF is calculated "locally" that is using an average over the specific simulation as reference. Second, when I look at Fig. S1 it appears that there are still some changes in the RMSF curves even towards the end of the simulations that are of the same magnitude (but in the opposite direction) as those observed during cooling. Is that correct or am I looking at the figure in the wrong way?

Also, while I realize that it is difficult to boil down a complex ensemble to one or a few numbers that can be tracked, it would be useful with alternative ways of looking at the ensembles. Are there local differences that are not captured by RMSF? What about rotamer distributions. I will leave it up to the authors whether to explore these issues further in this paper.

p. 11:

In terms of future experimental studies, what kinds of tests of the models could the authors envision? For example, the authors discuss work by Chen et al (Ref 24) on differences depending on the starting conditions. Do the authors' analytical model capture such effects? Do the authors' results lead to specific criteria for selecting good model systems to test the effects of cooling on conformational

ensembles?

p. 11/12:

Maybe the authors could also briefly discuss the relationship to other techniques that rely on (rapid) cooling including ssNMR and EPR. I realize that the cooling process is different, but it might still be worth speculating on how the approaches and models the authors present could be extended to other situations. In this context I'd also like to point out relevant work from Rob Tycko studying protein folding by ssNMR with rapid injection into a cold isopentane bath (<https://dx.doi.org/10.1021/ja908471n>).

Minor

p. 1/2: In the discussion of molecules settling into the lowest free energy minima at slow cooling rates, it might be worth making it clear that these minima may well be different from the minima at ambient temperatures.

p. 4: In the T-quenching MD simulations I couldn't easily find whether the simulations were performed using pressure control and if so how.

p. 6: "the atoms are subjected to harmonic potentials with a force constant c which are uniformly distributed in an interval from $-d$ to d " makes it sound like it is the force constants that are between $-d$ and d . Consider rephrasing.

p. 6 "Model3 is a combination of model2 and model3," should be, I guess, "Model3 is a combination of model1 and model2,"

p. 6: It is not clear what value of the pre-exponential factor that the authors use. I did not go through the maths, but I would assume that the choice would affect the "effective" barrier heights e.g. in Fig. 4. It would be useful if the authors would clarify this, given that there has/is some discussion about what pre-exponential factors are relevant for conformational changes in biomolecules.

p. 11: The authors write "Biomolecules can thermodynamically access more conformations at room temperature than at the cryogenic temperature". While that is probably mostly true, examples such as cold-denaturation suggest it isn't universally true.

I have discussed this review with Iris Young and James Fraser, UCSF, who have posted an independent review of the authors' preprint on bioRxiv. Their comments will be posted in the comments section at bioRxiv at <https://doi.org/10.1101/2021.10.08.463658>. I would recommend that the authors also examine the constructive and overall positive comments in that review, and note in particular a concordance with the point described above on boiling down complex properties of an ensemble to a few numbers.

Kresten Lindorff-Larsen, University of Copenhagen

Reviewer #2:

Remarks to the Author:

This paper concerns the effects of inevitable and essential quench-cooling of biomolecular samples for cryo-EM examination. With the growing recognition of, and interest in so-called structural heterogeneity, the topic is timely. Recent work on mapping structural variability has further

heightened the need for a better understanding of the effects of rapid cooling in cryo-EM sample preparation.

The authors combine an interesting set of theoretical techniques to examine and characterize the possible ways in which sample quenching can alter the conformational spectrum. The models presented in this paper involve many adjustable parameters. Fortunately, their results appear to show better agreement when the number of adjustable parameters is somewhat reduced. All in all, the models address a complicated question, and deserve serious attention.

The somewhat disappointing aspect of the work concerns the paucity of significant new insights from the carefully developed and described models. The primary conclusions are, in essence, what one would have guessed: sufficiently rapid cooling “freezes in” the conformational spectrum reminiscent of the uncooled spectrum, while slow cooling allows the system to relax to the minimum-energy conformation. It is left open whether the conformational spectra can be modeled in terms of a temperature at all.

In recognition of the seriousness and timeliness of the work, I recommend the authors consider the comments above, and also highlight the new insights, which warrant publication in *Nature Communications* rather than an excellent archival journal.

Reviewer #3:

Remarks to the Author:

This paper has discussed and quantified the generalized impact of plunge-freezing on native ensemble of biomolecules done to obtain EM images of the relevant population. The authors have performed extensive temperature quenching MD simulation of bacterial ribosome to investigate the contribution of cooling rate on the structural ensembles and reported that cooling-rate largely impacts equilibration into different local minima separated by 8 – 10 kJ/mol barrier heights. Authors have also reported that thermal contraction of the biomolecule due to cooling and equilibration within local minima are mostly cooling-rate independent effect of the freezing procedure. The robust discussion on the underlying physics of the plunge-freezing procedure might be useful for optimizing EM imaging process as required.

There are a few major concerns as listed:

1. RMSF is a commonly used metric for the amount of fluctuation occurring in a atom over time relative to its average position. However, RMSF spectrum fails to capture many significant yet subtle and small movements as they focus on the amount of change rather than the quality of change. As we know that majority of high frequency fluctuations correspond to thermal motions whereas moderate-to-low frequency fluctuations mostly captures the conformational rearrangements, it would be better if authors can comment on the effect of cooling-rate on different frequency ranges.

2. How cooling affects the dynamics of water (/water models) because proteins thermal fluctuation will also be influenced by the solvent’s dynamics?

There are a few minor concerns as listed below:

1. The RMSD of the predicted rmsf values from model-1 (Fig-3e), didn’t vary much even when trained with more simulation data. However, the RMSD value was consistently low while trained with all spans of simulation data, even with shorter spans (0.1 – 8ns). Can authors elaborate the trend?
2. For Q5, all the models have either overestimated or underestimated the rmsf while compared with simulation data correspond to all cooling-rates investigated. Any reason for why this quantile behaved

differently?

3. Page-6, line-223: "Model3 is combination of model2 and model3". Do authors mean model2 and model1?

4. Can authors elaborate what are "free parameters" and its implication?

Reviewer 1: Kresten Lindorff-Larsen

The manuscript by Bock & Grubmüller describes a detailed, multi-pronged computational study of the complex and important effects of cooling during sample preparation for cryo-EM. The paper is generally easy to read, appears technically sound and provides relatively clear results that will be of interest both to theoreticians and practitioners of cryo-EM.

Over the last 10 years cryo-EM has delivered increasingly high-resolution structures that in some cases now rival those of e.g. X-ray crystallography. In addition to examining the structures of macromolecules, cryo-EM may also provide more detailed insights into their energy landscapes because it in principle is a single-molecule technique that enables the visualization of the conformational distribution of molecules.

These advances leave two questions that have been difficult to answer. First, to what extent does the “average” structure under cryogenic conditions reflect the ambient temperature “average” structure [realizing that the term average here is somewhat misleading, the authors will understand what is meant]. Second, to what extent does the distribution of conformations (conformational ensemble) present in the cryo-EM sample reflect the distribution at ambient temperatures [leaving aside the technical difficulties of determining structural models of these ensembles from experiments]. While the first question can to a certain extent be answered by comparing structures solved at cryo-conditions with those at ambient temperature (by crystallography), the second question lies at the heart of the utility (and large potential) for cryo-EM to study conformational ensembles.

This study provides welcomed data in an area that has been lacking detailed and quantitative modelling, and where experiments are difficult. The results are promising in the sense that they support the idea that cryo-EM can to a large extent capture conformational ensembles at ambient temperatures. Importantly, the study provides a framework to think about these problems in a more quantitative manner that will hopefully spur additional experiments and analyses.

Specific comments:

Major

p. 4:

I must admit that I found the RMSF-based analysis somewhat difficult to follow in places. First, just to be sure could the authors confirm that in each case the RMSF is calculated

“locally” that is using an average over the specific simulation as reference. Second, when I look at Fig. S1 it appears that there are still some changes in the RMSF curves even towards the end of the simulations that are of the same magnitude (but in the opposite direction) as those observed during cooling. Is that correct or am I looking at the figure in the wrong way?

The understanding of the reviewer is correct. To calculate the rmsf values of an ensemble of structures, the structures were first aligned and then the rmsf is calculated with respect to the average structure of this ensemble. For example, the rmsf values before cooling were calculated from all 41 starting structures relative to the average of these 41 starting structures. The rmsf values after cooling were calculated from the final structures of 41 T-quench simulations with respect to their average. To clarify, we have extended the description of the rmsf calculation in the Results section:

“To that aim, we extracted structures of the ribosome complex, which contain all atoms resolved in the cryo-EM structure, at intervals of 50 ns from the 3.5- μ s simulation at 277.15 K (Fig. S1a). Then, we grouped the 41 structures from the time points between 0 μ s and 2 μ s into an ensemble, aligned the structures, and then calculated the rmsf of each atom with respect to the average structure of the ensemble.”

Indeed, the RMSF distributions are still changing with simulation time and are slowly increasing towards the end, which is, we agree, opposite of the expected cooling effect.

This at first sight non-intuitive behavior is discussed in section “Effects of different cooling rates on the ensemble” on page 4 of our original manuscript:

*“For later time intervals, the rmsf values increase with a decreasing slope (intervals 0.4–2.4 μ s to 1.5–3.5 μ s). The observed behavior is consistent with a slight adaptation of the initial ribosome structure obtained at cryogenic temperatures to near room temperature, with subsequent room temperature fluctuations. For the cooling simulations, we expected a decrease in rmsf values. To observe this effect, it is crucial to start cooling from an ensemble of structures which does **not** show a decrease in rmsf values in the absence of cooling. We therefore chose the ensemble of 41 structures in the 1–3 μ s interval (Fig. S1b, red histogram).”*

Unfortunately, there was a “not” missing (highlighted in red) which we believe created the confusion, understandably. It is now added.

However, the reviewer's comment also prompted us to have a closer look and providing further analyses of the following issue: Inevitably, any MD equilibration of any non-trivial biomolecular system will exhibit a slight remaining drift towards the equilibrium ensemble of the used force field, as often reflected by a slight increase of the usual rmsd with respect to the starting structure. This drift also contributes to the rmsf values. Taking an ensemble of

starting structures from the 277.15-K equilibration run, which under further sampling shows an increase in rmsf, results in a positive rmsf contribution also in the T-quench simulations, which might counteract the rmsf decrease due to cooling. This effect is particularly pronounced for slower cooling due to longer sampling. Overall, this partial cancellation can result in an underestimation of the cooling effect and, hence, the numbers we provided are actually conservative lower bounds.

To also provide an upper bound for this additional rmsf contribution, we have now in the revised manuscript measured the rmsf increase in the 277.15-K simulation for different time spans. As an example, consider the T-quench simulations with a cooling time span τ_c of 1 ns started from the 41 structures at times between 1 and 3 μ s (1000 ns, 1050 ns, ..., 3000 ns). To calculate how much the rmsf increases during an additional sampling of 1 ns at 277.15 K, we calculated the rmsf of 41 structures shifted by 1 ns (1001 ns, 1051 ns, ... , 3001 ns). We then subtracted this increase from the rmsf observed in the T-quench simulations. This procedure was repeated for all T-quench simulations and time points. The results are now shown in the newly added Fig. S8.

As expected, the resulting rmsf values (red) are generally lower than the rmsf values obtained from the T-quench simulations (black), but within the statistical uncertainty. Notably, because of the decreasing temperature in the T-quench simulations, the rmsf increase due to the additional sampling is expected to be smaller in the T-quench simulations compared to the 277.15-K simulations. Therefore, the rmsf values obtained from the subtraction (red) provide a lower limit of the expected rmsf decrease during cooling, thus now providing 'brackets' of upper (black curves) and lower limits (red), with the true values somewhere in between. To be on the conservative side, we used the T-quench values for all further analyses to make sure that we do not overestimate the effect of cooling.

We have now added a section to the Supplementary Results ("*Effect of additional sampling on rmsf during T-quench simulations*") to describe these results and added an explanatory sentence to the manuscript (p. 4):

"The small rmsf increase for later time intervals does not significantly affect the results of the T-quench simulations (for details, see Supplementary Results)."

Also, while I realize that it is difficult to boil down a complex ensemble to one or a few numbers that can be tracked, it would be useful with alternative ways of looking at the ensembles. Are there local differences that are not captured by RMSF? What about rotamer distributions. I will leave it up to the authors whether to explore these issues further in this paper.

We agree that describing the structural ensemble by only the distribution of the rmsf values of all atoms may provide a somewhat limited perspective. Clearly, spatial or chemical information as well as information on local relaxation motions will not be seen.

Motivated by the reviewer's comment, we now provide additional, orthogonal analyses in the revised manuscript:

To provide chemically resolved information, we divided the ribosome-complex atoms into subsets, one containing only RNA atoms, one comprising protein atoms, and separately calculated the rmsf distributions before and after the 128-ns T-quench simulations. Further, to provide spatially resolved information, we divided the set of atoms into five shells around the center of mass of the ribosome, where each set contains the same number of atoms. For each of these sets, we calculated rmsf distributions before and after the longest T-quench simulations with $\tau_c=128$ ns (Fig. S2).

Indeed, there are large differences in the absolute values of the rmsf quantiles, before and after cooling, between RNA and protein atoms with the protein atoms showing larger rmsf values (Fig. S2a). Also the atoms in the different shells around the center of the ribosome show different rmsf distributions. Mainly, as expected, the outer shells are more flexible than the inner shells.

However, and perhaps more surprisingly, the rmsf *reduction* during cooling, i.e. the difference between the rmsf before and after cooling, is remarkably similar for all the sets of atoms (Fig. S2b). This observation suggests that the rmsf distribution of all atoms provides, to a first approximation, a valid observable to capture the narrowing of the ensemble during cooling.

To also provide information on local relaxation motions, we now investigate in the revised manuscript to what extent the backbone conformations of the proteins are affected by the cooling rate. To this end, we extracted Φ and Ψ angles of the ensembles before cooling, as well as during and at the end of all the T-quench simulations. Ramachandran plots of the ensemble before cooling and at the end of the 128-ns T-quench simulation are now shown in the new Fig. S4a. Indeed, the distribution of angles narrows markedly during cooling. To quantify this effect, we calculated Shannon entropies for these distributions (Fig. S4b). The entropies decreased significantly during the T-quench simulations which nicely quantifies the narrowing of the protein backbone conformation distributions. Interestingly, slower cooling results in more pronounced entropy decreases, confirming the overall trend. This cooling-rate dependence suggests that some of the free-energy barriers which determine the cooling relaxation further analyzed below in the main manuscript actually involve backbone fluctuations and, likely, rearrangements.

We have carried out a similar analysis for rotamer angles χ_1 and χ_2 of the amino acids that contain these rotamers (Fig. S4c). As was seen for the backbone dihedrals, also their distributions become narrower during cooling, and this effect is also more pronounced with slower cooling.

We thank the reviewer for prompting these additional analyses which in our view provide valuable insights into the shock freezing relaxation dynamics of biomolecules. We have now added a paragraph to the Results section to discuss these results (p. 6).

p. 11:

In terms of future experimental studies, what kinds of tests of the models could the authors envision? For example, the authors discuss work by Chen et al (Ref 24) on differences depending on the starting conditions. Do the authors' analytical model capture such effects? Do the authors' results lead to specific criteria for selecting good model systems to test the effects of cooling on conformational ensembles?

Our kinetic model does not capture the scenario that the relative free-energy difference between conformational states can change with the temperature. We assume that this is probably underlying the dependence of the trapped conformation on the starting conditions in the work by Chen et al. Our simulations also do not capture a situation like this, because the ensemble, from which the T-quench simulations were started, is generated only at one temperature. Therefore, such a model cannot be validated against our simulation data.

In our view, a good test of this assumption would be to generate ensembles at several temperatures, check if the (Boltzmann corrected) free-energy differences between conformational states differ for different temperatures, and finally start T-quench simulations from these ensembles to see how well the ratios of the conformational states are preserved during cooling.

A good test system would have relatively few states, separated by free-energy barriers that can be overcome during MD simulations. Further, the free energy of some of the states should be dominated by enthalpy while that of others should be entropic, such that a strong dependency on temperature would be seen. We think that such follow-up studies are out of the scope of the present manuscript.

We have now added a paragraph to the discussion:

"We hypothesize that the dependence of the trapped conformation on the temperature prior to cooling observed in some cryo-EM experiments^{24,25,26} might be due to temperature dependent free-energy differences between conformational states. Although our MD simulations fully capture how free energies change with temperature and even the associated non-equilibrium effects, we here started all simulations from an ensemble at one (ambient) temperature, such that this effect cannot be explored. For this reason, we have also not included temperature dependent free-energy differences between conformational states within our simple kinetic models. One might test this hypothesis by generating ensembles at several temperatures, then starting T-quench simulations from these ensembles, and quantifying the occupancy of conformational states during cooling."

p. 11/12:

Maybe the authors could also briefly discuss the relationship to other techniques that rely on (rapid) cooling including ssNMR and EPR. I realize that the cooling process is different, but

it might still be worth speculating on how the approaches and models the authors present could be extended to other situations. In this context I'd also like to point out relevant work from Rob Tycko studying protein folding by ssNMR with rapid injection into a cold isopentane bath (<https://dx.doi.org/10.1021%2Fja908471n>).

Good idea! As suggested, we have now added the following sentence and references to the introduction:

“Rapid cooling, with the freeze-quench method²⁷, is also used in electron paramagnetic resonance (EPR) spectroscopy experiments to trap intermediate states²⁸ and in combination with solid-state NMR experiments allows the identification of transient folding intermediates²⁹.”

We did not expand too much on this subject, because the main focus of our study is clearly on the interpretation of cryo-EM data.

Minor

p. 1/2: In the discussion of molecules settling into the lowest free energy minima at slow cooling rates, it might be worth making it clear that these minima may well be different from the minima at ambient temperatures.

We fully agree, and have now changed the sentence in the introduction discussing the work of Chen et al., to make clear that the minima can depend on the temperature. We also added a reference to Singh et al. *Nat Struc Mol Biol* 2019 (as suggested by reviewers Iris Young and James Fraser):

“The observation that captured conformations of a ketol-acid reductoisomerase and of temperature-sensitive TRP channels differ dramatically for different temperatures prior to cooling suggests that, in these cases, the minimal free-energy conformations depend on the temperature and that the conformations are preserved during rapid cooling^{24,25,26}.”

p. 4: In the T-quenching MD simulations I couldn't easily find whether the simulations were performed using pressure control and if so how.

Thanks for pointing out the missing information; we have now added a sentence to the Methods Section:

“The pressure was coupled to a Parrinello-Rahman barostat⁸⁸ ($T_p = 1$ ps).

p. 6: “the atoms are subjected to harmonic potentials with a force constant c which are uniformly distributed in an interval from $-d$ to d ” makes it sound like it is the force constants that are between $-d$ and d . Consider rephrasing.

We have changed the sentence to "In this model, the atoms are subjected to harmonic potentials with a force constant c . The minima of the potentials are uniformly distributed in an interval from $-d$ to d (Fig. 3)."

p. 6 "Model3 is a combination of model2 and model3," should be, I guess, "Model3 is a combination of model1 and model2,"

Yes, that was indeed a typo and is now corrected. Thanks!

p. 6: It is not clear what value of the pre-exponential factor that the authors use. I did not go through the maths, but I would assume that the choice would affect the "effective" barrier heights e.g. in Fig. 4. It would be useful if the authors would clarify this, given that there has/is some discussion about what pre-exponential factors are relevant for conformational changes in biomolecules.

We used a temperature-dependent pre-exponential factor $\kappa\left(\frac{T}{T_h}\right)^\nu$, where T is the current temperature and T_h is the temperature prior to cooling. For the temperature exponent ν , we tested values 0.5, 1, and 2. The choice of ν did not affect how well the kinetic models agree with the MD simulation data (Figure 3e) and we used 1 for the further analysis.

As assumed by the reviewer the choice of the scaling factor κ indeed affects the barrier height. Therefore, for model3, we tested values of κ between 0.01 ns^{-1} and 10 ns^{-1} , obtained the model parameters with Metropolis sampling and calculated the deviation between rmsf values obtained from the MD simulations and the rmsf values obtained from the model (we have now added Supplementary Fig. S9). The deviation decreases with increasing κ values and stays constant when 1 ns^{-1} is reached, suggesting that $\kappa \geq 1 \text{ ns}^{-1}$. Since larger values of κ result in larger barrier heights, our results using a κ value of 1 ns^{-1} provide a lower limit of the barrier height. We have now added these results to the Supplementary Information (section "Pre-exponential factor for model2 and model3").

p. 11: The authors write "Biomolecules can thermodynamically access more conformations at room temperature than at the cryogenic temperature". While that is probably mostly true, examples such as cold-denaturation suggest it isn't universally true.

We agree that this statement, as written, is not universally true, as cold denaturation shows. However, we are not aware of any case where cold denaturation has compromised cryo-EM structure determination, presumably because upon plunge-freezing the glass-transition temperature is reached on timescales shorter than those typically expected for unfolding, as our calculations also have shown. We note that any potential or partial cooling-induced unfolding events would also show up in our simulations. We have rephrased the respective sentence on p. 11 to make this issue clearer.

“Generally, biomolecules thermodynamically access more conformations at room temperature than at the cryogenic temperature and rates of conformational changes are determined by free-energy barriers and temperatures²². For very rapid cooling, low temperatures that prevent barrier crossing are quickly reached and the conformations prior to cooling are expected to be kinetically trapped. In contrast, during very slow cooling, most barriers are expected to be overcome and low free-energy conformations are predominately occupied resulting in an ensemble that is more homogeneous than the room-temperature ensemble prior to cooling. However, to what extent the rapid cooling perturbs the ensembles is currently unknown.”

Further down in the discussion (p. 12), we have added a paragraph with a reference to Dias et al. *Cryobiology* 2010 (reference 766):

“We note that during cold denaturation a transient increase of the accessible number of conformations can occur during cooling⁷⁶, which, however, is not relevant to cryo-EM studies, because during plunge-freezing the biomolecule typically reaches the glass-transition temperature on timescales much shorter than those expected for unfolding. In our simulations, we have not observed unfolding events either, which provides additional support for this notion.”

I have discussed this review with Iris Young and James Fraser, UCSF, who have posted an independent review of the authors' preprint on bioRxiv. Their comments will be posted in the comments section at bioRxiv at <https://doi.org/10.1101/2021.10.08.463658>. I would recommend that the authors also examine the constructive and overall positive comments in that review, and note in particular a concordance with the point described above on boiling down complex properties of an ensemble to a few numbers.

We were glad to see the review by Iris Young and James Fraser on bioRxiv and addressed all their comments at the end of this reply letter (reviewers 4)

Kresten Lindorff-Larsen, University of Copenhagen

Reviewer 2

This paper concerns the effects of inevitable and essential quench-cooling of biomolecular samples for cryo-EM examination. With the growing recognition of, and interest in so-called structural heterogeneity, the topic is timely. Recent work on mapping structural variability has further heightened the need for a better understanding of the effects of rapid cooling in cryo-EM sample preparation.

The authors combine an interesting set of theoretical techniques to examine and characterize the possible ways in which sample quenching can alter the conformational spectrum. The models presented in this paper involve many adjustable parameters. Fortunately, their results appear to show better agreement when the number of adjustable parameters is somewhat reduced. All in all, the models address a complicated question, and deserve serious attention.

We appreciate the positive and encouraging comments of the reviewer. Indeed, as the reviewer points out, the finding that out of the two cooling-rate dependent models we have studied, the one with the least number of adjustable parameters (model 3) reproduced and predicted the rmsf quantiles derived from the atomistic cooling simulations best is important, as it clearly speaks against possible overfitting. As the wording of the reviewer's comment might be read as suggesting a remaining possibility of overfitting, we would like to summarize here why we are convinced that overfitting is not an issue here. Generally, it is not the mere number of adjustable parameters that points to overfitting, but their ratio to the number of independent data points to which they are fitted. Here, our kinetic model 3 reproduced and predicted 605 rmsf values, involving only 9 free parameters (while other model variants required up to 25 parameters). Clearly, not all of these 605 rmsf values are fully statistically independent, but because the model is actually able to describe the evolution of rmsf values over four orders of magnitude in cooling time spans (0.1–128 ns), we think the reviewer will agree that these data contain way more information than actually needed for a stable fit. As an additional guard against overfitting, we performed careful cross-validation by excluding parts of the data from the Bayes fitting (Fig. 3e) and to test different sets of parameters (Fig. S5), akin to calculating free R factors in X-ray crystallography.

The somewhat disappointing aspect of the work concerns the paucity of significant new insights from the carefully developed and described models. The primary conclusions are, in essence, what one would have guessed: sufficiently rapid cooling “freezes in” the conformational spectrum reminiscent of the uncooled spectrum, while slow cooling allows the system to relax to the minimum-energy conformation.

In light of the appreciation of our new insights by all other reviewers, this comment was a bit of a surprise. We were actually quite happy to see that our simulations confirm the qualitative expectation that rapid cooling prevents the equilibration into lower free-energy minima – had they not, our manuscript would have been rightly questioned due to conflict with experimental data. However, as appreciated by reviewers 1 and 3 as well as by Young and Fraser, our approach provides new quantitative data on how fast the cooling proceeds, how much, for given cooling speed, the ambient temperature ensemble narrows, as well as a new quantitative relationship between the heights of the barriers that are overcome during cooling and the temperature drop during cooling.

This relationship informs the interpretation of cryo-EM experiments and quantifies the limitations of current cooling procedures. In an increasing amount of cryo-EM experiments, multiple distinct conformations are routinely resolved and our results suggest that these conformations are separated by barriers larger than ~ 10 kJ/mol with current cooling protocols. To resolve conformations separated by lower barriers, more rapid cooling would be required. Besides this cooling-rate dependent effect, we quantified the cooling-rate independent effects of equilibration within local minima and thermal contraction.

We are confident that our gained detailed understanding of how structural ensembles are affected by the cooling rate will inspire experiments where both the temperature prior to cooling and the cooling rate are varied. Then, analysis of the observed occupancies of different conformations based on our framework has the potential to quantify not only the free-energy differences between conformations at room temperature but also the heights of separating barriers.

We have re-written parts of the discussion (p. 11-13) to better highlight these novel findings.

It is left open whether the conformational spectra can be modeled in terms of a temperature at all.

We have tried hard to understand this comment, but have to admit that we are still not quite sure if we fully captured its meaning. In case we misunderstood this comment, we would very much welcome a clarification and the opportunity to reply.

One way we read the reviewer's comment is to ask whether or not there is a one-to-one relationship between the (width of?) conformational spectra and temperature, i.e., that the temperature uniquely determines the conformational ensembles. But because our simulations clearly show that the cooling process is far from equilibrium, the answer is 'no' and, hence, not left open as claimed.

We therefore think it is more likely that the reviewer criticized lack of detail in our characterisation of how the conformational ensembles change with temperature. To address this concern, we have now obtained the dominant conformational modes and checked if the distribution along these modes is affected by the cooling. To that end, we first carried out a principal component analysis (PCA) of structures extracted at 5 ps intervals from the trajectory at $T=271.15$ K (1-3 μ s). To calculate the covariance matrix, we used the coordinates

of the P and CA atoms of RNA strands and proteins, respectively. The eigenvalues of each eigenvector of the covariance matrix correspond to the variance of the projections onto this eigenvector, such that eigenvectors with large eigenvalues describe dominant conformational modes of motion.

To restrict the analysis to those conformational modes which contribute most to the overall motion, we chose the 212 eigenvectors whose eigenvalues are larger than 1 Å for further analysis. Next, for each eigenvector, we projected the structures from which the T-quench simulations were started on the eigenvectors and calculated the standard deviation of the projections σ_{before} . For each cooling time span τ_c , we then projected the structures at the end of the T-quench simulations and calculated the standard deviation σ_{after} . The normalized difference between the standard deviations $\Delta\sigma=(\sigma_{\text{after}}-\sigma_{\text{before}})/\sigma_{\text{before}}$ for the eigenvectors as a function of the eigenvalues is now shown in Fig. S10.

For rapid conformational changes, one would expect the distributions of projections to become narrower upon cooling ($\Delta\sigma<0$) and that this effect is more pronounced for longer cooling time spans. We do not see a clear trend of narrowing, which could be explained by the conformational changes being too slow or a consequence of limited statistics.

For the more local changes measured by rmsf as well as the narrowing of backbone dihedral angles and side-chain rotamers, however, we see a clear effect of the cooling (see answer to comment by reviewer 1). Together, these results suggest that the local changes converge faster than the correlated motions involving the whole ribosome, which provides additional information on the structural determinants underlying the different barrier heights derived from model 3.

We now describe these new results in an additional section in the Supplement "*Effect of cooling on conformational modes*".

It would be interesting to see if with more statistics a narrowing is also observed for distribution of the conformations along the conformational modes. However, testing this on the large ribosome complex is computationally not feasible. It would be better to carry out extensive simulations on a small protein with more rapid conformational modes instead. This, however, would be a project on its own and is outside of the scope of the presented manuscript.

In recognition of the seriousness and timeliness of the work, I recommend the authors consider the comments above, and also highlight the new insights, which warrant publication in Nature Communications rather than an excellent archival journal.

We appreciate this recognition and have followed the recommendation as described in reply to the reviewer's comment above.

Reviewer 3

This paper has discussed and quantified the generalized impact of plunge-freezing on native ensemble of biomolecules done to obtain EM images of the relevant population. The authors have performed extensive temperature quenching MD simulation of bacterial ribosome to investigate the contribution of cooling rate on the structural ensembles and reported that cooling-rate largely impacts equilibration into different local minima separated by 8 – 10 kJ/mol barrier heights. Authors have also reported that thermal contraction of the biomolecule due to cooling and equilibration within local minima are mostly cooling-rate independent effect of the freezing procedure. The robust discussion on the underlying physics of the plunge-freezing procedure might be useful for optimizing EM imaging process as required.

We thank the reviewer for this positive assessment.

There are a few major concerns as listed:

1. RMSF is a commonly used metric for the amount of fluctuation occurring in a atom over time relative to its average position. However, RMSF spectrum fails to capture many significant yet subtle and small movements as they focus on the amount of change rather than the quality of change. As we know that majority of high frequency fluctuations correspond to thermal motions whereas moderate-to-low frequency fluctuations mostly captures the conformational rearrangements, it would be better if authors can comment on the effect of cooling-rate on different frequency ranges.

We thank the reviewer for this comment. This was in fact also suggested by reviewers 1 and 4, so we have carefully selected several new metrics, which we have now analyzed and included within the revised manuscript . For our detailed reply, please see our response to reviewer 1, where we discuss that effects of cooling are indeed seen for different kinds of motion which are occurring on different time scales. Briefly, the distributions of the protein side-chain rotamers and of the protein backbone dihedrals were observed to become narrower in a cooling-rate dependent manner, as observed for the rmsf.

As detailed in our response to reviewer 2, we have now also included an analysis of the correlated motions of the ribosome complex obtained using principal component analysis (PCA). Interestingly, and in contrast to the above metrics, we do not see a clear trend of narrowing for the distributions of these large conformational changes. An explanation of this finding could either be that the motions are too slow to be equilibrated during cooling, or limited statistics. Indeed, from cryo-EM experiments (e.g. Fischer et al. 2013) we know that several large-scale motions, such as the intersubunit rotation, are too slow to be equilibrated during cooling.

However, the amount of simulations required to distinguish between these possibilities, is currently computationally not feasible. In future work, extensive simulations of a small protein with rapid conformational changes could help to address this question.

2. How cooling affects the dynamics of water (/water models) because proteins thermal fluctuation will also be influenced by the solvent's dynamics?

We certainly agree that the biomolecular fluctuations are strongly affected by the solvent and, hence, this is clearly an important question. Note, however, that (at least for smaller proteins) the coupling between solvent and protein dynamics at different temperatures, in particular in relation to the protein glass transition, has already been extensively investigated using neutron scattering experiments (Réat et al. *PNAS* 2000), MD simulations with different water models (Steinbach et al. *PNAS* 1993, Norberg et al. *PNAS* 1996, Vitkup et al. *Nat Struct Biol* 2000, Tournier et al. *Biophys J* 2003), and X-ray crystallography (Sartor et al. *J Phys Chem* 1993, Tilton et al. *Biochemistry* 1992, Warkentin et al. *J Appl Crystallogr* 2009). There are also two reviews on this topic (Ringe et al. *Biophys Chem* 2003, Doster *Biochim Biophys Acta Proteins Proteom* 2010). All these papers are cited in our introduction. To better point the reader to this literature, we have now changed the sentence introducing these papers to “The effects of low temperatures on protein dynamics and the coupling between solvent and protein dynamics have been studied extensively using...”

Based on the extensive discussion in the literature and to keep the focus of the manuscript on protein/RNA dynamics, we would prefer not to include an analysis of water dynamics within our manuscript.

There are a few minor concerns as listed below:

1. The RMSD of the predicted rmsf values from model-1 (Fig-3e), didn't vary much even when trained with more simulation data. However, the RMSD value was consistently low while trained with all spans of simulation data, even with shorter spans (0.1 – 8ns). Can authors elaborate the trend?

Thank you for this observation. We have now added a sentence to the results (p. 8) to discuss this observation:

“The deviations are similar, because the parameter distributions obtained from the different training sets were similar, suggesting that the contribution to the rmsf decrease described by this model, namely the equilibration in local harmonic potentials, is indeed cooling-rate independent.”

2. For Q5, all the models have either overestimated or underestimated the rmsf while compared with simulation data correspond to all cooling-rates investigated. Any reason for why this quantile behaved differently?

We thank the reviewer for pointing this out. We have added the following paragraph on page 8:

“All models either underestimate or overestimate the median values of the rmsf quantile Q5. However, the rmsf values obtained from the models lie very well inside the confidence intervals (Fig. 3d, gray area). Q5 quantifies the rmsf values of the atoms with the largest heterogeneity in the ensemble. We expect the underlying large conformational changes to be slower than conformational changes resulting in smaller rmsf values. Therefore, we expect the conformational changes underlying Q5 to be less equilibrated, which is supported by the larger confidence intervals.”

3. Page-6, line-223: “Model3 is combination of model2 and model3”. Do authors mean model2 and model1?

Thank you, that was indeed a typo and is now corrected.

4. Can authors elaborate what are “free parameters” and its implication?

We assume the reviewer suggested a more detailed description of precisely which free parameters we have chosen, which we now have added to the manuscript (see below). But because the reviewer’s question might also be read as what we mean by the term “free parameters”, and to be sure we are talking about the same thing, let us first provide an explanation: Free parameters are the ‘unknowns’ if you wish, model parameters that enter the likelihood function; their probability distribution is calculated (see Fig. S6) by applying the Bayes theorem and Metropolis sampling of the probabilities. This probability distribution yields their expectation values and uncertainty estimates. We have adjusted the text to make this clear (p. 8):

“The free model parameters optimized by the Bayes approach are d and c for model1, ΔG^\ddagger , Δx , ΔG for model2, as well as d , c , ΔG^\ddagger , ΔG , and Δx for model3.”

Reviewers 4: Iris Young and James Fraser (via biorxiv)

The capability of cryo-electron microscopy (cryoEM) to capture multiple and native-like conformations of large macromolecules is transforming structural biology. This manuscript explores intricacies of the cooling process as it relates to structural ensembles. Specifically, how do variations in starting sample conditions (water layer thickness, water/sample starting temperature) and cooling (ethane layer thickness, rate of cooling) affect the distribution of structural states captured in the resulting micrographs? Can we be confident that the results of "time-resolved" cryoEM experiments are representative of barriers and basins we hope to capture? By a combination of molecular dynamics simulations and cryoEM experiments, the authors guide us to an empirical understanding of these questions.

To understand the relationship between cooling and structural ensembles, we must begin with the thermodynamic principles in play. Ensembles represent the many possible conformational states of a structural unit, and the occupancies of the individual states depend on the energy landscape across which they are related. At the extremes, we may imagine an ensemble cooled instantaneously to 0 K, whose component structures would not be able to traverse the energy landscape in any direction, as well as an ensemble injected with enough energy to overcome any energy barrier on the landscape (i.e. a system at thermal equilibrium), whose component structures would move freely to occupy all possible states. In the latter case we would not expect all states to be occupied by the same number of particles, however — particles with exactly enough energy to breach a particular energy barrier are equally likely to fall to either side of it, but particles at different starting points with the same starting energy have different likelihoods of escaping their local energy minima. In aggregate, this produces the Boltzmann distribution, in which the populations of different states depend entirely on their relative energies.

For intermediate temperatures, it is useful to speak in shorthand of energy barriers and a system's ability to overcome them. We believe the introduction of this manuscript deserves a more complete illustration of the fundamental principles, however, and would encourage the authors to possibly even add a diagram or two to aid in this effort. It is too easy to confuse the fact that cooled structures move toward global energy minima with the idea that cooling gives them the energy needed to overcome energy barriers, which is precisely the opposite of the truth. The abstract in particular ought to be reworded to avoid this misinterpretation.

As suggested by the reviewers, we added a schematic of how cooling may affect ensembles (Fig. 1a). We also improved the abstract and introduction text to make it clearer that it is the additional time spent at higher temperatures that allows for barrier crossing into lower free-energy minima.

The design of the computational and wet lab experiments is carefully geared toward isolating the relevant variables and reproducing the relevant states and processes. For

example, to choose the equilibration times to use in MD simulations, the authors first simulated how long it would take for samples of various thicknesses to vitrify. By and large we are satisfied with the parameters of these experiments, but we question one unsupported assumption: the authors enforced an ethane bath outer boundary held at constant temperature. This could be possible if the ethane bath remains in contact with a heat sink and the equilibration time between ethane and the heat sink is negligible compared with that between ethane and the sample, but we do not see this discussed or justified, nor is this standard practice when freezing grids, as the ethane bath is isolated from the standard liquid nitrogen heat sink after reaching the desired temperature to prevent the ethane from freezing solid. We were confused by the plot of equilibration times for ethane layers of different thicknesses, and hypothesize that the unexpected (to us) trend is a result of this boundary condition: we would have imagined a thicker ethane layer to allow quicker absorption of heat from the water layer, but the opposite is shown. Moreover, there is often a cold gas layer (see work by Rob Thorne on hyperquenching: <https://pubmed.ncbi.nlm.nih...> and <https://journals.iucr.org/m...>). While this complication might be very difficult to simulate, it should be explained how it might affect the interpretation of results.

Concerning the wet lab experiments mentioned in the first sentence of the reviewer's comment, we think there was a misunderstanding: We did not perform any wet lab experiments, and what the reviewer is likely referring to (Fig. 1b, we assume) are continuum solutions of the heat equation for the described system. Upon re-reading the abstract, we realized that unclear phrasing in the abstract might have caused the misunderstanding. We have now changed the abstract accordingly.

Concerning the motivation for the thickness and the heat bath coupling of the ethane bath, we would like to clarify that in our work we have used three layers of modeling. (1) estimation of the temperature drops during plunge-freezing (solution of the heat equation), (2) calculation of the response of a macromolecular system to different cooling rates (MD simulations of ribosome complex), and, (3) several kinetic models trained against the MD simulation data and then applied to the temperature drops estimated by model layer (1).

We have now expanded the respective summary of these three layers in the Discussion of our revised manuscript to avoid any misunderstanding.

As pointed out by the reviewers, for the estimation of the temperature drop the temperature at the outer boundaries, which are in contact with the ethane, is indeed kept constant. In contrast to what the reviewers seem to have assumed, however, we have not chosen to model the walls of the ethane container as close as possible to the experimental situation – which is different, as the reviewers correctly point out. The reason for this choice is that in the real experiments the ethane container is much larger than any reasonable grid size of the numerical continuum calculation would allow us to implement. To solve the heat

equation, however, a boundary condition is nevertheless necessary, which raises the question to which extent the too narrow ethane bath affects the results.

Because of the small water-layer width, low temperatures (< glass transition temperature) in the water layer are expected to be reached well before the temperature increase in the ethane layer reaches the walls of the ethane container and we therefore do not need to include the whole ethane container in the model. It suffices to make the ethane layer wide enough such that the temperature drop in the water layer is not affected by the ethane-layer width.

The various chosen ethane layer thicknesses aim to answer this question and help to extrapolate to the real situation of essentially 'infinite' thickness (for which the boundary condition becomes irrelevant). In particular, to test how wide the ethane layer should be, we successively increased the width (100 nm up to 3.2 μm). As can be seen, the difference between the temperature drops for widths 1.6 μm and 3.2 μm is very small and only occurs after around 10^{-5} s which is much longer than the time it takes to reach the glass transition temperature (< 10^{-6} s). We therefore chose the temperature drops obtained for 3.2- μm width for the following analysis.

This reasoning was not clear enough from the text, and we have now modified it accordingly.

We also thank the reviewers for pointing us to the papers by Robert Thorne and we have now included the following into our discussion:

"Above the liquid cryogen, a cold gas layer with a thickness of several mm was observed [Warkentin et al. 2006, Engstrom et al. 2021]. To what extent the cryo-EM samples are already cooled when they move through the gas layer during plunging is not clear. The slower temperature drop due to precooling by the cold gas layer, would allow biomolecules to overcome higher barriers resulting in more homogeneous ensembles."

Although it does not impact the methods or results of this paper, we are also unconvinced that the use of any vitrification bath held at a lower temperature than the commonly used ethane bath would necessarily result in faster freezing, as heat transfer is also dependent on heat capacity (hence the selection of ethane for plunge-freezing rather than liquid nitrogen!) Propane does indeed appear to have a higher heat capacity than ethane at similar cryogenic temperatures, so in the case of an ethane-propane mixture, this assumption does hold, but we would prefer the authors include this detail.

We agree with this assessment, and in the interest of staying within the word limit of Nature Communications, we have decided to remove this part, which is not essential for our main conclusions.

As for the results of the study, it is well-evidenced and clearly presented that conformational distributions present before plunge-freezing are reflected in vitrified samples when a

standard vitrification protocol is followed, and that the rate of cooling does indeed impact the degree to which these distributions are preserved. We especially applaud the authors' careful wording around what interpretations are supported or suggested by the data, leaving open the remote possibility of other explanations — they draw very clear distinctions between observations and analyses. This is good science!

Finally, we find the implications of the study meaningful. The selection of a ribosomal complex as an example particle perfectly illustrates the biologically relevant range of flexibilities and temperature-dependent conformational ensembles. This example gives us an intuitive measuring stick for other types of structures. Taken as a whole, the analyses inform future "time-resolved" studies using cryoEM and the design of other experiments that depend on the preservation of a conformational ensemble by rapid cooling. This is a very exciting direction of inquiry that we will continue to watch with great interest!

We thank the reviewer for these kind words.

Minor points:

- The authors could make the introduction even more clear and accessible by specifying that liquid specimens present a challenge because their vapor pressure is incompatible with high vacuum.

We have changed the corresponding sentence to: *"In general, biomolecules perform their functions in solution. However, the direct study of specimens in liquid solutions using EM is impeded because the high vacuum required by EM is incompatible with the vapor pressure of liquid solutions."*

- We favor the wording "most often liquid ethane" over "mostly liquid ethane" for describing the standard vitrification setup.

We agree and the sentence is now changed.

- Sentences such as the second to last sentence in the second paragraph of the introduction could be broken into multiple sentences or otherwise simplified to avoid confusion among the several "which," "from" and "and" clauses.

We have now carefully gone over the revised manuscript to avoid too long sentences and ambiguous 'which' etc references.

- Sobolevsky's work on TRP channels and vitrification probably deserves a mention in the intro alongside the other examples, esp. because those probe a natural temperature sensor! (<https://www.nature.com/arti...>, <https://www.nature.com/arti...>)

We thank the reviewers for pointing us to this amazing work. We have added both references to the introduction:

“The observation that captured conformations of a ketol-acid reductoisomerase and of temperature-sensitive TRP channels differ dramatically for different temperatures prior to cooling suggests that, in these cases, the minimal free-energy conformations depend on the temperature and that the conformations are preserved during rapid cooling^{24,25,26}.”

- Tomography can be used to resolve position in ice layer, “Apart from the water-layer thickness, the temperature drop also depends on the position within the layer with the slowest drop in the center (Fig. 1a), which is relevant, because in the time between the spreading of the sample onto the grid and the plunging, the biomolecules tend to adsorb to the air-water interface 62.” (<https://pubmed.ncbi.nlm.nih...>)

The introduction already referred to this work; we have now added an additional reference at this position.

- The authors could comment on whether the effects of being in different parts of the ice layer would affect RMSF values. If the effects are not too small, reconstructions from different layers could yield B-factors that could deconvolute different effects!

We thank the reviewers for this nice idea and have added the following to the discussion: *“Finally, we would expect cooling rates to depend on the position in the ice layer, e.g., depending on the distance to the surface of the sample layer. Reconstructions from different positions could further help to disentangle the effects.”*

- Are there local effects to their RMSF kinetic/thermodynamic models in the ribosome? For example, could they subdivide RNA/Protein, small/large subunit, exterior/interior sites, etc? Are there any regions that increase conformational heterogeneity upon cooling (as we have seen often in multi temperature crystallography)? Using a global RMSF metric may be leaving out interesting phenomena.

This has also been a suggestion by reviewers 1 and 3, which led us to include a number of additional analyses to this effect. Please see our reply to the comment of reviewer 1 for a detailed description.

- Are the frames and code deposited somewhere for others to examine?

We will make the structures of the ensembles before and after cooling available for download (zenodo.org), with a link provided in the Data Availability statement. The code can be obtained from the authors upon request.

Iris Young and James Fraser (UCSF)

Reviewers' Comments:

Reviewer #1:

Remarks to the Author:

I thank the reviewers for their detailed response and additional analyses. I in particular think the new analyses of dihedral angle changes strengthened the paper. Indeed, the revisions have overall made an already very nice paper even better. Congratulations.

Kresten Lindorff-Larsen

Reviewer #2:

Remarks to the Author:

The authors have made sufficiently dealt with my comments. In my view, the paper can be accepted.

Reviewer #3:

Remarks to the Author:

The authors have addressed all the points raised during the review process and the manuscript is ready for publication

Reviewer #4:

Remarks to the Author:

The authors have more than adequately responded to our comments here: <http://disq.us/p/2mq2u86> and we find their responses enlightening. They should endeavor to release as much analysis code as they can!